# Revealing intact neuronal circuitry in centimeter-sized formalin-fixed paraffin-embedded brain

Ya-Hui Lin[1,2], Li-Wen Wang[1,2], Yen-Hui Chen[3], Yi-Chieh Chan[1], Shang-Hsiu Hu[1], Sheng-Yan Wu[1], Chi-Shiun Chiang[1], Guan-Jie Huang[4], Shang-Da Yang[5], Shi-Wei Chu[4], Kuo-Chuan Wang[6], Chin-Hsien Lin[7], Pei-Hsin Huang[8], Hwai-Jong Cheng[9], Bi-Chang Chen[10]*, Li-An Chu[1,2]*

[1]Department of Biomedical Engineering and Environmental Sciences, National Tsing Hua University, Hsinchu, Taiwan; [2]Brain Research Center, National Tsing Hua University, Hsinchu, Taiwan; [3]Institute of Biomedical Sciences, Academia Sinica, Taipei, Taiwan; [4]Department of Physics, National Taiwan University, Taipei, Taiwan; [5]Institute of Photonics Technologies, National Tsing Hua University, Hsinchu, Taiwan; [6]Department of Neurosurgery, National Taiwan University Hospital, Taipei, Taiwan; [7]Department of Neurosurgery, National Taiwan University Hospital, Taipei, Taiwan; [8]Department of Pathology, National Taiwan University Hospital, Taipei, Taiwan; [9]Institute of Molecular Biology, Academia Sinica, Taipei, Taiwan; [10]Research Center for Applied Sciences, Academia Sinica, Taipei, Taiwan

*For correspondence:
chenb10@gate.sinica.edu.tw (B-ChangC);
lachu@mx.nthu.edu.tw (L-AnC)

Competing interest: The authors declare that no competing interests exist.

## Abstract

Tissue-clearing and labeling techniques have revolutionized brain-wide imaging and analysis, yet their application to clinical formalin-fixed paraffin-embedded (FFPE) blocks remains challenging. We introduce HIF-Clear, a novel method for efficiently clearing and labeling centimeter-thick FFPE specimens using elevated temperature and concentrated detergents. HIF-Clear with multi-round immunolabeling reveals neuron circuitry regulating multiple neurotransmitter systems in a whole FFPE mouse brain and is able to be used as the evaluation of disease treatment efficiency. HIF-Clear also supports expansion microscopy and can be performed on a non-sectioned 15-year-old FFPE specimen, as well as a 3-month formalin-fixed mouse brain. Thus, HIF-Clear represents a feasible approach for researching archived FFPE specimens for future neuroscientific and 3D neuropathological analyses.

## eLife assessment

The reprocessing and reanalysis of archived samples can yield further insights from past experiments. Here, a **useful** procedure to perform tissue clearing and immunolabeling on large-scale formalin-fixed paraffin-embedded brain specimens is **convincingly** evaluated on a set of archival pathology specimens, and its applicability to further such samples is analyzed. This method will be of interest to both neuroscientists and pathologists.

## Introduction

Clinical brain tissues play a critical role in unraveling the molecular mechanisms underlying neurological diseases and facilitating the development of effective treatments (*Rajput and Rajput, 2015*; *Hernandez-Ronquillo et al., 2020*; *Latimer et al., 2023*), and also represent an invaluable resource

for retrospective studies on rare diseases or conditions with long-term outcomes.*Talari and Goyal, 2020*. Currently, the majority of archived clinical specimens are stored in the form of formalin-fixed paraffin-embedded (FFPE) blocks. Advanced molecular analysis tools—such as chromatin immunoprecipitation, DNA/RNA sequencing, and Luminex assays—have been performed on FFPE specimens to enable comprehensive analysis from genomic, transcriptional, and proteomic perspectives (*Latimer et al., 2023*; *Jacobsen et al., 2023*; *Zhao et al., 2021*; *Zhong et al., 2019*). However, image-based qualification and quantification of FFPE samples remain limited to performing chemical staining and immunohistochemistry on thin sections. Although whole-slide imaging enables unbiased analyses of entire tissue sections (*Latimer et al., 2023*; *Fraggetta et al., 2018*; *Kumar et al., 2020*; *Patel et al., 2021*), its two-dimensional (2D) nature cannot provide the 3D information required for accurate quantification. Such quantification in neurobiological contexts is crucial for evaluating neuronal damage and neuroprotection, including neuronal cell counts and the extent of necrosis. Moreover, such 2D-based techniques also fail to reveal structural changes within target tissues.

Tissue-clearing protocols (*Richardson and Lichtman, 2015*; *Ueda et al., 2020*; *Gómez-Gaviro et al., 2020*; *Richardson et al., 2021*) in combination with deep-tissue immunolabeling (*Yun et al., 2019*; *Susaki et al., 2020*; *Yau et al., 2023*) and optical sectioning microscopy techniques (*Conchello and Lichtman, 2005*; *Dodt et al., 2007*; *Reynaud et al., 2008*) represent powerful tools for collecting 3D information at a subcellular resolution from centimeter-scale specimens. Tissue-clearing techniques render biological tissues transparent by removing components that induce refraction index mismatching, which is mainly attributable to two factors, that is, light absorption due to pigments or biological chromophores and light scattering mainly caused by lipids (*Richardson and Lichtman, 2015*; *Yu et al., 2021*). In current tissue-clearing protocols, samples are fixed with 4% paraformaldehyde (PFA) for ~24 hr before undergoing delipidation steps that involve solvents (*Ertürk et al., 2012*; *Renier et al., 2014*; *Jing et al., 2018*), aqueous solutions with denaturing reagents or detergents (*Hama et al., 2011*; *Susaki et al., 2014*; *Hama et al., 2015*; *Tainaka et al., 2018*), or protection with chemicals that create crosslinks to withstand harsh electrophoretic delipidation (*Chung et al., 2013*; *Murray et al., 2015*; *Park et al., 2019*). Such protocols now enable imaging of whole-rodent bodies (*Cai et al., 2019*; *Mai et al., 2024*) and even entire human organs at the cellular or subcellular level (*Zhao et al., 2020*), facilitating detailed and comprehensive analyses of intact biological systems. However, FFPE specimens have not reaped the benefits of these advanced techniques. For instance, although tissue-clearing approaches such as CUBIC, iDISCO, CLARITY, and passive Sodium dodecyl sulfate (SDS) treatment have been applied previously to FFPE specimens (*Chen et al., 2019*; *Hughes et al., 2014*; *Lai et al., 2018*; *Nojima et al., 2017*; *Tanaka et al., 2017*; *Tanaka et al., 2018*), they have only yielded immunolabeling depths of 1–2 mm, underscoring the need for a procedure tailored to FFPE samples in order to achieve centimeter-scale tissue clearing and labeling.

Our objective was to develop a protocol capable of rendering centimeter-scale FFPE specimens optically transparent and suitable for deep immunolabeling. To this end, we took account of two critical issues. First, successful immunohistochemistry of FFPE tissue sections requires antigen retrieval to restore target conformation and expose masked epitopes (*Shi et al., 2001*; *Thavarajah et al., 2012*). Secondly, despite xylene treatment during paraffin processing and dewaxing protocols removing a substantial proportion of FFPE tissue lipids, certain phospholipids persist within the lipid bilayer, impeding antibody penetration and maintaining opacity (*Nojima et al., 2017*; *Denti et al., 2020*). Therefore, we aimed to devise a procedure that achieves efficient antigen retrieval and delipidation for centimeter-scale FFPE specimens. Here, we present heat-induced FFPE-based tissue clearing (HIF-Clear), a pipeline that employs an optimized heat-induced antigen retrieval technique to achieve that goal. We validated the effectiveness of HIF-Clear using FFPE human brain tissues and entire mouse brains. Using HIF-Clear, we demonstrate the possibility of conducting multiple rounds of immunolabeling, which allowed characterization of the same brain tissue using six different neuronal markers and revealing neuronal circuitry. We further demonstrate volumetric quantification by means of HIF-Clear on intact FFPE mouse brains, with statistical analysis of the FFPE samples resulting in identical trends to the results determined for fresh brains. We also showcase the broad applications of HIF-Clear by successfully conducting volumetric imaging of mouse brain archived in FFPE blocks for a period of 15 y, as well as the compatibility of this approach with expansion microscopy.

## Results

### Optimized antigen retrieval: Delipidation and epitope recovery of FFPE specimens

SDS is a detergent commonly used for lipid removal in tissue-clearing methods. It is also known as an antigen retrieval agent and has been used to enhance immunolabeling signals in cells (*Robinson and Vandré, 2001*), cryosections (*Brown et al., 1996*), formalin-fixed human brains (*Woelfle et al., 2023*), and epithelia of the pancreas, liver, and lung (*Messal et al., 2021*). Therefore, we began optimizing tissue clearing by applying two types of SDS-based approaches, passive and active, to FFPE mouse brain hemispheres and then evaluated their impacts on antigen retrieval/delipidation by means of electrophoretic immunolabeling (*Yun et al., 2019*). For the passive approach, we applied a FLASH protocol published previously involving sample incubation in an SDS-based solution (4% SDS and 200 mM borate) at 54°C for antigen retrieval and tissue clearing of epithelial cells and abdominal organs (*Messal et al., 2021*). For the active approach, we employed SDS-based electrophoresis at 42°C for 48 hr. We observed that both FLASH-processed and SDS-electrophoresed samples showed weak tyrosine hydroxylase (TH, a marker of dopaminergic neurons) signal (*Figure 1—figure supplement 1a*). Additionally, we noticed that the FLASH-processed samples had almost no signal of NeuN (a marker of neuronal nuclei, *Figure 1—figure supplement 1b*, left), and exhibited strong nonspecific background noise (*Figure 1—figure supplement 1a*, left). The presence of this background noise is considered an indicator of inadequate antigen retrieval (*Kim et al., 2016*).

We found that FLASH-processed samples displayed weak TH (a marker of dopaminergic neurons) signal and almost completely lacked signal of NeuN (a marker of neuronal nuclei) (*Figure 1—figure supplement 1a, left, and b, left*). We also observed that the SDS-electrophoresed sample only possessed weak TH-positive signal, as well as strong nonspecific labeling (*Figure 1—figure supplement 1a*, right), with this latter considered an indicator of inadequate antigen retrieval (*Kim et al., 2016*).

These results imply insufficient antigen retrieval using the SDS approaches, likely due to the low-temperature experimental conditions. In 1991, Shi et al. published a heat-induced epitope retrieval (HIER) method utilizing a temperature of 100°C (*Shi et al., 1991*), which now represents the most broad-spectrum antigen retrieval protocol for FFPE tissue sections. Superheating (i.e., temperatures >100°C) in pressure cookers has been reported to enhance immunohistochemistry signal intensity and stability (*Norton et al., 1994*; *Taylor et al., 1996*). However, heating with a classical antigen retrieval citrate buffer (10 mM sodium citrate, pH 6.0) at 121°C for 10 min in a pressure cooker, a condition appropriate for routine FFPE sections, only resulted in weak immunostaining signals of NeuN and SMI312 (a pan-neurofilament marker) in the middle of a 2-mm-thick FFPE mouse brain block (*Figure 1—figure supplement 1b*, middle column). Since dewaxed and rehydrated FFPE specimens still have residual phospholipids that prevent antibody penetration (*Nojima et al., 2017*; *Denti et al., 2020*), we added 1% SDS into the citrate buffer to increase the delipidation capability of the HIER approach and found that NeuN and SMI312 signals were significantly enhanced (*Figure 1—figure supplement 1b*, right). However, deploying SDS at such high temperatures also resulted in tissue expansion and fragility (*Figure 1—figure supplement 1b*, gross view), prompting us to search for a detergent condition that could achieve sufficient delipidation while retaining tissue integrity.

Accordingly, we tested four additional detergent conditions: 1% Tween 20, 1% Triton X-100, 1% sodium cholate (NaC), and 1% CHAPS. Tween 20 and Triton X-100 are commonly used to disrupt cell membranes in immunohistochemistry. NaC and CHAPS have higher critical micelle concentrations and form smaller micelles than SDS, and both have been reported as being more efficient than SDS with respect to active electrophoretic delipidation (*Na et al., 2022*) and passive delipidation (*Zhao et al., 2020*), respectively. All tests were performed on 2-mm-thick FFPE mouse brain blocks with citrate buffer (10 mM sodium citrate, pH 6.0) at 121°C for 10 min in a pressure cooker. First, we assessed amounts of residual lipids by staining with DiD (a lipophilic dye with an excitation wavelength at 649 nm). Undelipidated and SDS-delipidated (hereafter referred to as PFA-SDS) mouse brain blocks, as well as dewaxed FFPE mouse brain blocks without HIER, were also stained as controls. To prove that DiD signal quantification could be used to reflect amounts of residual lipids in tissue sections, we compared DiD signal across the FFPE group without HIER, the undelipidated group and the PFA-SDS group (*Figure 1A*, left panel). The FFPE group without HIER exhibited a higher DiD signal intensity than the PFA-SDS group but lower than that of the undelipidated group, supporting partial lipid

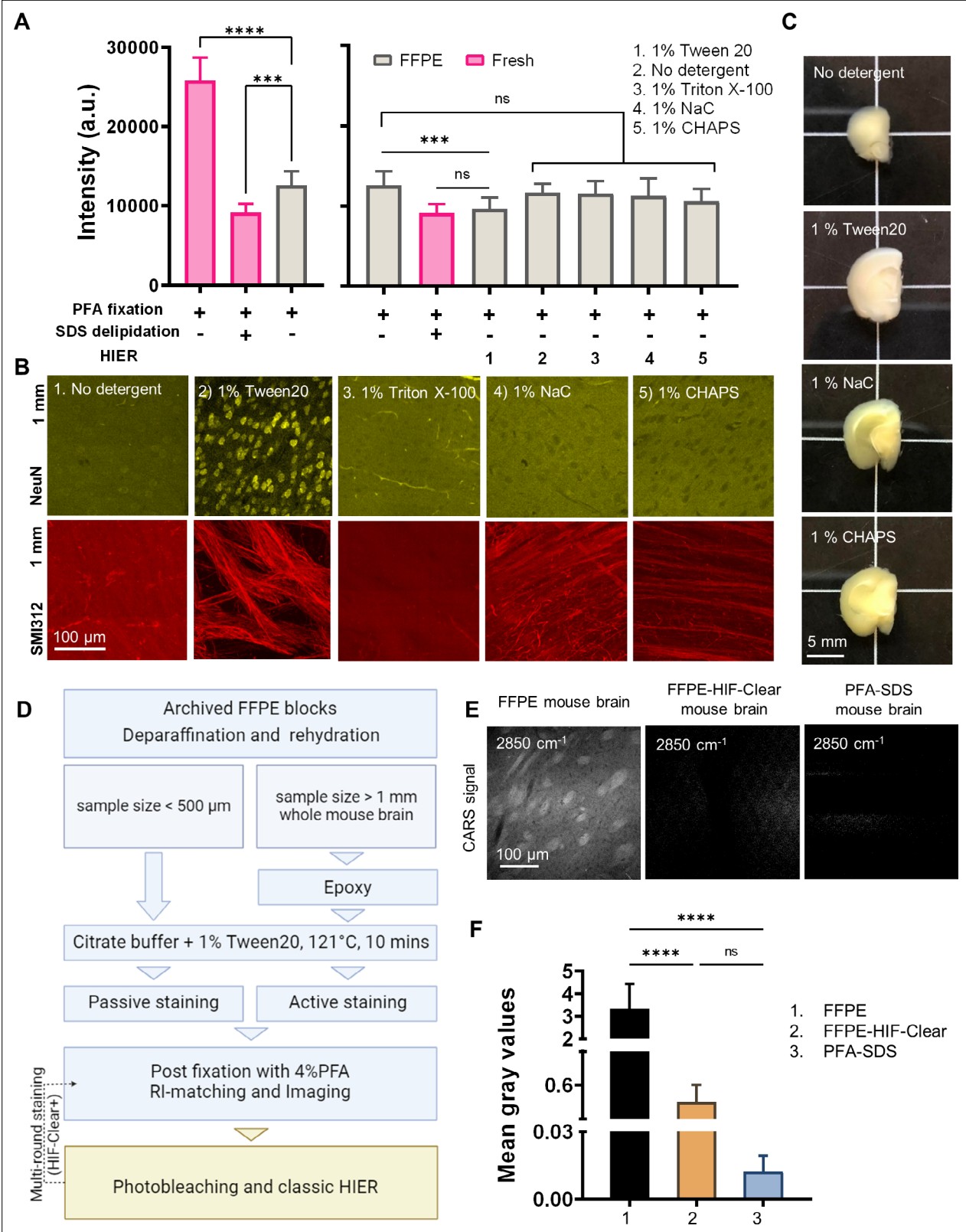

**Figure 1.** Development of the HIF-Clear pipeline. (**A**) Comparison of DiD signal intensity in 2-mm-thick slices of undelipidated formaldehyde-fixed mouse brain, PFA-SDS mouse brain, and formalin-fixed paraffin-embedded (FFPE) mouse brain treated under various detergent conditions. (**B**) Multi-point confocal fluorescence images of neuronal nuclei in the cortex (top row) and axons in the striatum (bottom row) from 2-mm-thick FFPE mouse brain slices that had been treated under various detergent conditions. Images of optical sections captured at a depth of 1 mm are shown. Detergent types

*Figure 1 continued on next page*

*Figure 1 continued*

and concentrations are indicated. NeuN, a neuronal nuclear marker; SMI312, a pan-axonal marker. (**C**) Gross views of 2-mm-thick slices of FFPE mouse brain treated under various detergent conditions. (**D**) Schematic of the HIF-Clear pipeline. (**E**) Images of coherent anti-Stokes Raman scattering (CARS) of dewaxed FFPE mouse brain, HIF-Clear-processed FFPE mouse brain, and PFA-SDS mouse brain. The images were taken at a Raman shift of 2850 cm$^{-1}$ to address the CH2 vibration frequency. (**F**) Statistical analysis of the gray values for the three treatment groups in (**E**). Gray values for 18 z-continuous images of each group were analyzed. Statistical analysis: (**A**) n = 10; (**E**) n = 18. Mean ± SD, *p<0.05, **p<0.01, ***p<0.001, ****p<0.0001, one-way ANOVA with Tukey's multiple comparison test.

The online version of this article includes the following figure supplement(s) for figure 1:

**Figure supplement 1.** Examination of the antigen retrieval effect of SDS on formalin-fixed paraffin-embedded (FFPE) specimens.

**Figure supplement 2.** Representative images of the results of DiD and Oil red O staining.

**Figure supplement 3.** Repeated dewaxing test.

**Figure supplement 4.** Repeated dewaxing does not influence antigenicity.

**Figure supplement 5.** Gross view of the mouse brain at different stages of the HIF-Clear protocol.

removal during the paraffin processing and dewaxing process. Among all of the test groups subjected to varied heated detergent conditions, only the 1% Tween 20 condition presented significantly lower lipid content compared to the FFPE without HIER group (p-value=0.0035), achieving the same level of delipidation as the PFA-SDS group (p-value=0.999) (*Figure 1A*, right panel). We also performed Oil red O staining to visualize the residual lipids and tissue morphology under bright-field microscopy (*Figure 1—figure supplement 2*).

Next, we evaluated the efficiency of antigen retrieval by conducting immunolabeling with NeuN and SMI312 antibodies on all of the test groups and examined the signals at the center of each FFPE specimen. We found that the 1% Tween 20 group displayed the sharpest and clearest neuronal nuclei and axon bundles with a high signal-to-background ratio, indicating both successful delipidation and antigen retrieval (*Figure 1B*, panel 2). Interestingly, for the 1% Triton X-100, 1% NaC, and 1% CHAPS groups, the fluorescence intensity of neuronal nuclei was even lower than the background signal, implying that heating these detergents might hinder nuclear epitopes (*Figure 1B*, panels 3–5). We also examined the gross appearance of each group upon heating, which confirmed that the 1% Tween 20 group retained good sample integrity without apparent changes in size or color (*Figure 1C*). Therefore, we employed 1% Tween 20 in citrate buffer with heating in a pressure cooker at 121°C for 10 min as the critical step of our HIF-Clear pipeline for delipidation and antigen retrieval of centimeter-scale FFPE specimens.

## The HIF-Clear pipeline

We illustrate the complete HIF-Clear pipeline in *Figure 1D*. To retrieve a specimen from an FFPE block, the paraffin is first melted at 65–70°C, before dewaxing at room temperature using xylene and sequentially rehydrating the tissue with alcohol. It is important to note that when working on a large sample, the dewaxing time should be increased to avoid opacity caused by residual wax due to insufficient dewaxing. Opacity can be eliminated through repeated dehydration and dewaxing steps (*Figure 1—figure supplement 3b*), with extended dewaxing times or repeated dewaxing steps not affecting tissue structure or antigenicity (*Figure 1—figure supplements 3 and 4*). A typical dewaxing timeframe for paraffin sections of less than 20 µm thickness is ~20 min, with complete dewaxing of an entire mouse brain or similarly sized sample achievable within 24 hr (see 'Materials and methods). Optimal dewaxing times for tissue sizes between these two extremes warrant further testing.

The rehydrated FFPE samples are then treated with epoxy (*Park et al., 2019*) and immersed in the optimized antigen retrieval solution (10 mM sodium citrate, 1% Tween 20, pH 6.0) at 37°C for 24 hr, before then being heated to 121°C for 10 min. The samples are then subjected to active immunolabeling (see 'Materials and methods'). After staining, the specimen is immersed in 4% PFA for 24 hr to fix the antibodies bound to epitopes, matched for the refractive index (RI), and imaged by means of light-sheet microscopy. *Figure 1—figure supplement 5* shows the gross views of the same mouse brain after undergoing 4% PFA fixation, paraffin processing, optimized antigen retrieval, and RI matching, demonstrating intactness of the brain shape and preservation of tissue integrity. To enable 3D phenotyping and proteomic investigations, a multi-round staining protocol involving photobleaching and classic heat-induced antigen retrieval can also be included. In this study, unless

otherwise specified, all HIF-Clear-processed whole mouse brains had been stored in paraffin blocks for 2–6 mo before the experiments and underwent active labeling after heating with 1% Tween 20.

To further verify HIF-Clear's efficacy in removing lipids from whole mouse brains, we employed coherent anti-Stokes Raman scattering (CARS), a spectroscopic technique for detecting vibrational responses of specific molecular structures (*Figure 1E*). In contrast to a dewaxed FFPE mouse brain (*Figure 1D*, left) for which we observed CARS signals from axon bundles, thus indicating the presence of myelinated lipids, both the FFPE-HIF-Clear-processed and PFA-SDS-treated mouse brain samples displayed significantly lower CARS signals (*Figure 1E*, middle and right, respectively; quantification in *Figure 1F*), confirming removal of myelinated lipids.

## The HIF-Clear pipeline enables immunolabeling and registration of intact FFPE mouse brain

Previous studies have confirmed that transcriptomic and proteomic data are equally preserved in FFPE sections and fresh-frozen sections. To assess biomarker preservation within a spatial context, we investigated patterns of anti-TH antibody labeling in FFPE-HIF-Clear-processed mouse brains (*Figure 2A*). We observed consistent distributions of dopaminergic neurons in key tissue regions, including the striatum, nigrostriatal fiber tracts, substantia nigra, ventral tegmental area, pontine reticular nucleus, and lateral reticular nucleus, aligning with the findings of a prior study (*Susaki et al., 2020*; *Figure 2A*, left) and matching results from Allen Brain Atlas registration (*Claudi et al., 2020*; *Figure 2A*, right). Furthermore, we compared the volume of each brain region between the left and right hemispheres, which revealed minimal sample distortion (*Figure 2B*). Our investigation extended to a 2-mm-thick mouse brain that had been preserved in an FFPE block for 15 y. By applying HIF-Clear, we unveiled clear clusters of dopaminergic neurons at a cellular resolution in that specimen (*Figure 2C*). These results confirm the feasibility of deploying HIF-Clear for long-term stored FFPE blocks, even without the need for sectioning, enabling effective retrieval of information on spatial biomarkers comparable to that obtained from freshly fixed samples. We also applied HIF-Clear to 3-month fixed mouse brain hemispheres (*Figure 3*). Although the long-term fixed specimens exhibited reduced TH intensity and S/N ratio, the major dopaminergic regions were labeled, and magnified images revealed clear details of cell bodies and neuronal fibers. These results suggest that HIF-Clear has the potential to be applied to long-term fixed specimens.

## A tailored HIF-Clear+ pipeline enables multi-round immunolabeling of human brain tissues and whole mouse brains

Next, we investigated whether HIF-Clear-processed whole-brain FFPE samples could be subjected to a multi-round staining process, a strategy commonly used for multiplexed immunolabeling. Chemical elution techniques utilizing detergents, reducing reagents, and denaturing reagents have been developed previously and tested on thin paraffin sections or cryosections. For thick sections, the CLARITY protocol published in 2013 involves incubation in 4% SDS solution, which was first demonstrated on a 1-mm-thick mouse hippocampus specimen. However, based on our preliminary experiments, we found that post-staining fixation is required to maintain signals in a centimeter-sized sample (i.e., a whole mouse brain) (*Figure 3—figure supplement 1*) and the post-fixed antibodies cannot be completely stripped using SDS-based stripping protocols (*Figure 3—figure supplement 2*). We also observed that fixation after staining hindered subsequent rounds of staining, although this impediment could be overcome by traditional heat-induced epitope retrieval (referred to hereafter as classical HIER to avoid confusion with the aforementioned optimized antigen retrieval protocol) (*Figure 3—figure supplement 3*). To overcome the difficulty of antibody stripping due to post-fixation, which is required for centimeter-sized specimens, we developed HIF-Clear+ to include a multi-round immunolabeling protocol utilizing photobleaching and classical HIER (*Figure 1D*). After imaging, we photobleached transparent RI-matched samples using a 100 W LED white light to quench the previously labeled fluorophores (*Figure 3—figure supplement 4*). The selected duration of bleaching is dependent on sample thickness; overnight bleaching is sufficient for tissue samples of <500 μm thickness, whereas ~72 hr are required for thicker samples such as whole mouse brains (a thickness of 0.8–1 cm). Bleached specimens are then subjected to classical HIER to restore the antigenicity hindered by post-staining PFA fixation. The brain shape and structural integrity remained after four rounds of immunolabeling, and

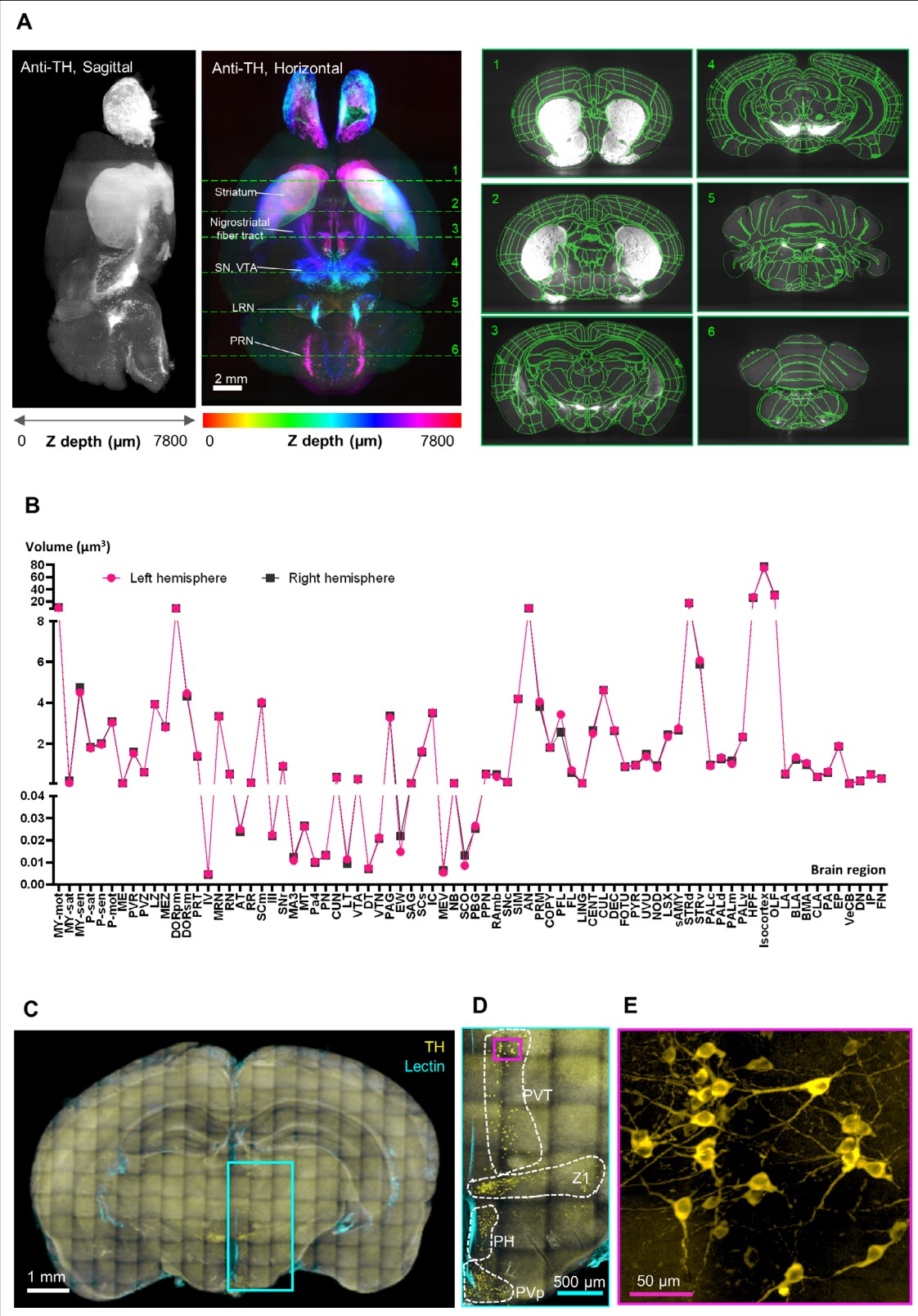

**Figure 2.** Validation of the HIF-Clear pipeline. (**A**) Brain-wide tyrosine hydroxylase (TH) expression (left panel) and registration (right panel) in FFPE-HIF-Clear C57BL/6 mouse brains. Whole mouse brain images were acquired using light-sheet microscopy, and the 3D-rendered images of the horizontal (XY) and sagittal (YZ) views are shown. Major dopaminergic regions are indicated: SN (substantia nigra), VTA (ventral tegmental area), LRN (lateral reticular nucleus), and PRN (pontine reticular nucleus). The results of brain registration for these regions (marked with green dotted lines and

*Figure 2 continued on next page*

*Figure 2 continued*

numbered in the XY view) are displayed from the coronal (XZ) view in the right panel. (**B**) Comparison of brain region volumes between the left and right hemispheres of an FFPE-HIF–Clear mouse brain. All abbreviations are listed in ***Supplementary file 2***. (**C–E**) Application of HIF-Clear to a 15-year-old FFPE block with TH immunolabeling and multi-point confocal imaging. Specimen thickness: 2 mm. (**C**) The projection image. (**D**) Magnification of the cyan-lined region in (**C**). Dopaminergic regions are indicated. PVT, paraventricular nucleus of thalamus; PH, posterior hypothalamic nucleus; PVp, periventricular hypothalamic nucleus. (**E**) Magnification of the magenta-lined region in (**D**). Scale bars are indicated in each panel. FFPE, formalin-fixed paraffin-embedded.

there is no cross-reactivity in subsequent rounds of immunostaining following bleaching (***Figure 3— figure supplement 5***).

To explore the potential of revealing multiple neuronal circuitry in the whole FFPE mouse brain using HIF-Clear+, we performed a six-round immunolabeling procedure for TH, tryptophan hydroxylase 2 (TPH2), calbindin, parvalbumin (PV), choline acetyltransferase (ChAT), and SMI312, respectively, on one intact FFPE mouse brain (***Figure 4***). For each round, the brain was labeled not only with the antibody specific for the target protein, but also for lectin as a structural reference. Image datasets from each immunolabeling round were combined into a single multichannel image (***Figure 5A*** and ***Figure 4—video 1***), with lectin-labeled blood vessels deployed as a reference channel using Elastix (***Klein et al., 2010***), demonstrating the ability of HIF-Clear+ to reveal the spatial distribution of different neuronal markers. We conducted a more detailed examination of the habenula-interpeduncular circuitry (Hbn-IPN circuitry) and its connection with these biomarkers. We visualized the expression of ChAT, calbindin, TH, and TPH2 in the IPN and the neighboring regions (***Figure 5B***). Previous studies have shown that these neurons are regulated by the Hbn-IPN circuitry through their connections with the IPN region, which plays a role in controlling emotion, addiction, and the reward process (***McLaughlin et al., 2017***; ***Fakhoury, 2018***). To demonstrate that HIF-Clear+ can also be applied to human brain FFPE blocks, we performed three consecutive rounds of immunolabeling on a 1-mm-thick FFPE human brain specimen collected surgically from the temporal lobe of a brain tumor patient. For each round, the brain tissue was labeled with GFAP/lectin, SMI312/lectin, or MAP2/lectin, respectively (***Figure 6A***). Labeling homogeneity throughout the entire block was confirmed by imaging at depths of 100, 500 and 900 μm (***Figure 6B***, ***Figure 6—figure supplement 1***).

## Application of FFPE-HIF-Clear to mouse disease models

In disease research, volumetric imaging offers precise microenvironmental information and reveals structural changes that are essential for evaluating injuries and therapeutic effects (***Gómez-Gaviro et al., 2020***; ***Chen et al., 2019***; ***Liebmann et al., 2016***). As a first assessment of HIF-Clear applicability to disease models, we subjected an intact FFPE brain from an astrocytoma mouse model (see 'Materials and methods') to the HIF-Clear pipeline to label tumor cells (ASTS1Cl-GFP positive astrocytes) and GFAP-positive astrocytes (***Figure 7—video 1***, ***Figure 7A and C***). Accordingly, we could segment GFAP-positive astrocytes surrounding the tumor (***Figure 7B, D and E***) and classify them according to their distances from the tumor cells. Statistical analysis (***Figure 7F***) revealed that nearly half of the GFAP-positive astrocytes were within the tumor, with 63.9% being located near the tumor surface (±200 μm). These results demonstrate that FFPE-HIF-Clear-derived image datasets can reveal cell distributions and the spatial architecture of disease microenvironments.

Next, we employed FFPE mouse brains from a traumatic brain injury (TBI) model to showcase the feasibility of employing HIF-Clear for 3D quantification of neuronal circuit recovery in disease research. Reduced dopamine levels are known to be a major cause of cognitive impairment after TBI (***Jenkins et al., 2018***). To assess the impact on dopaminergic neural regeneration across four treatment groups (sham, sMN, aNORO@sMN, and aNORO@sMN+AMF; see 'Materials and methods' for details), we conducted immunolabeling using anti-TH antibodies (***Figure 7G***) and compared quantitative changes in the dopaminergic system among these groups. We calculated quantitative ipsilateral-to-contralateral changes for three structural features: striatum volume (***Figure 7H***), nigrostriatal fiber tract volume (***Figure 7I***), and the number of dopaminergic cells in the substantia nigra (SN) (***Figure 7J***). The sham group displayed a decrease (~24%) in the ratio of dopaminergic neurons on the injured side to the uninjured control side (***Figure 7K***), indicative of dopaminergic system malfunction or imbalance. Treatment with aNORO@sMN rescued this disease phenotype (from 76% to 94%) (***Figure 7K***), a positive impact on dopamine production and transmission recovery post-TBI.

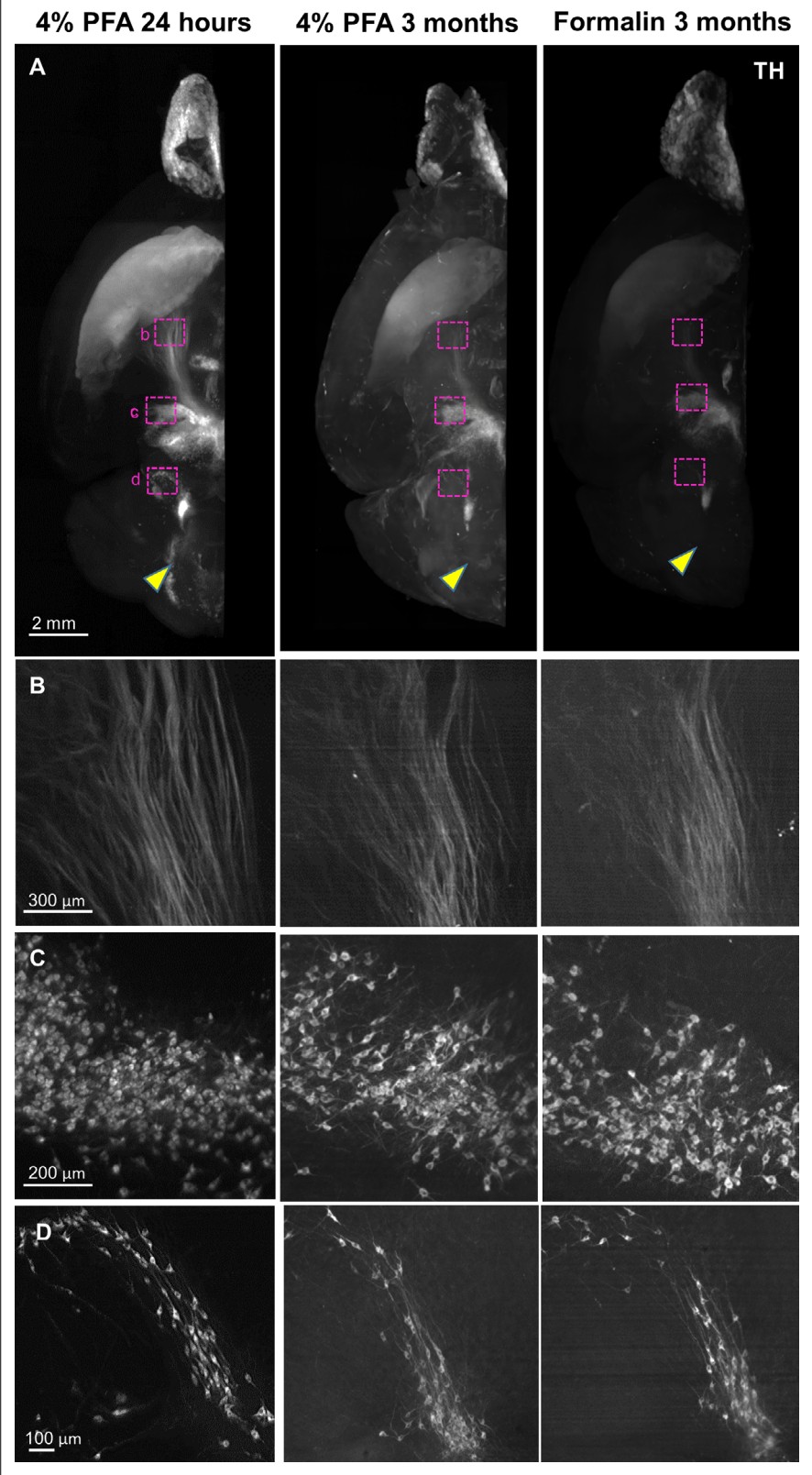

**Figure 3.** Application of HIF-Clear to long-term fixed mouse brain hemispheres. Mouse brain hemispheres were fixed under various fixation conditions and then subjected to formalin-fixed paraffin-embedded (FFPE) processing and HIF-Clear. The specimens were stained with tyrosine hydroxylase (TH) antibodies and imaged with light-sheet microscopy. (**A**) Projection images of the horizontal (XY) view. The magenta-lined regions indicating nigrostriatal

*Figure 3 continued on next page*

*Figure 3 continued*

fiber tract, ventral tegmental area (VTA), and lateral reticular nucleus (LRN) are magnified in (**B**), (**C**), and (**D**), respectively. The yellow arrowheads indicate pontine reticular nucleus (PRN), which was clearly immunolabeled in the 4% paraformaldehyde (PFA) 24-hour-fixed specimen but not in the 3-month-fixed specimens.

The online version of this article includes the following figure supplement(s) for figure 3:

**Figure supplement 1.** Immunolabeled whole mouse brain without post-fixation exhibits faint or negative staining.

**Figure supplement 2.** Post-immunolabeling-fixed tyrosine hydroxylase (TH) signal remains after SDS stripping.

**Figure supplement 3.** Classic heat-induced epitope retrieval (HIER) facilitates relabeling after post-immunolabeling fixation.

**Figure supplement 4.** Photobleaching in HIF-Clear+.

**Figure supplement 5.** Examination of issue structural integrity and secondary antibody cross-reactivity of HIF-Clear+.

---

Next, we evaluated the therapeutic effect of aNORO@sMN on angiogenesis. To quantify angiogenesis, we selected three regions of interest (ROIs) around brain injury sites and segmented blood vessels (*Figure 7L* and *Figure 7—video 2*). The aNORO@sMN treatment group exhibited a substantial increase in blood vessel length, surface area, volume, and branch numbers (244, 301, 186, and 220%, respectively) compared to the sham group (*Figure 7M*), indicating enhanced angiogenesis. Thus, overall, HIF-Clear facilitates the 3D quantification critical for assessing neuronal damage and recovery in FFPE specimens.

We also performed identical TBI procedures and 3D quantification using the SHIELD protocol and SDS-based electrophoresis for brain clearing and labeling (*Figure 7—figure supplement 1*), which resulted in comparable quantitative outcomes compared to those determined for FFPE-HIF-Clear brains (*Figure 7—figure supplement 2*). These results indicate that FFPE-HIF-Clear recovers biomarker signals as effectively as for freshly processed samples, even after the samples have been subjected to prolonged storage as FFPE blocks.

## HIF-Clear enables expansion microscopy on thick FFPE samples

Expansion microscopy is a super-resolution microscopy technique that overcomes the limitations of optical diffraction by enlarging biological structures. It is achieved by synthesizing a polymer network within the specimen to anchor protein molecules, causing the specimen to physically expand and enabling analysis of sub-cellular structures (*Chen et al., 2015*; *Tillberg et al., 2016*). It is generally difficult to uniformly expand heavily formaldehyde-fixed specimens, such as FFPE tissues, due to the formaldehyde occupying the binding sites of anchoring reagents. ExPath, a modified version of protein-extension expansion microscopy (ProExM) in which the EDTA concentration in the proteinase K digestion buffer is increased, was developed to overcome this issue (*Zhao et al., 2017*). However, although ExPath has been successfully deployed to expand 5-µm-thick FFPE sections, it cannot uniformly expand samples beyond 1 mm even with a prolonged incubation time (*Figure 8A*). We discovered that tissues embedded in paraffin and subjected to the optimized antigen retrieval step in the HIF-Clear pipeline can be directly expanded using the original ProExM method without extended incubation, yielding clear staining signal for abundant targets such as SMI312 or MAP2 (*Figure 8B and C*).

## Discussion

In this study, we propose HIF-Clear, which utilizes a heat-induced antigen retrieval protocol optimized with 1% Tween 20, to eliminate residual lipids and recover masked antigens from FFPE specimens. Our method enables labeling agents to penetrate to a depth of centimeters within such specimens, enabling large-scale neuronal circuitry analyses of archived brain tissues and providing comprehensive information that previously could not typically be obtained from conventional 3–5 µm FFPE sections. We have demonstrated herein that FFPE samples subjected to the HIF-Clear pipeline can be repeatedly stained without compromising their integrity. Moreover, in evaluating therapeutic efficacy for a TBI mouse model, we found that HIF-Clear-treated FFPE mouse brains provided as much statistical information as shortly PFA-fixed mouse brains.

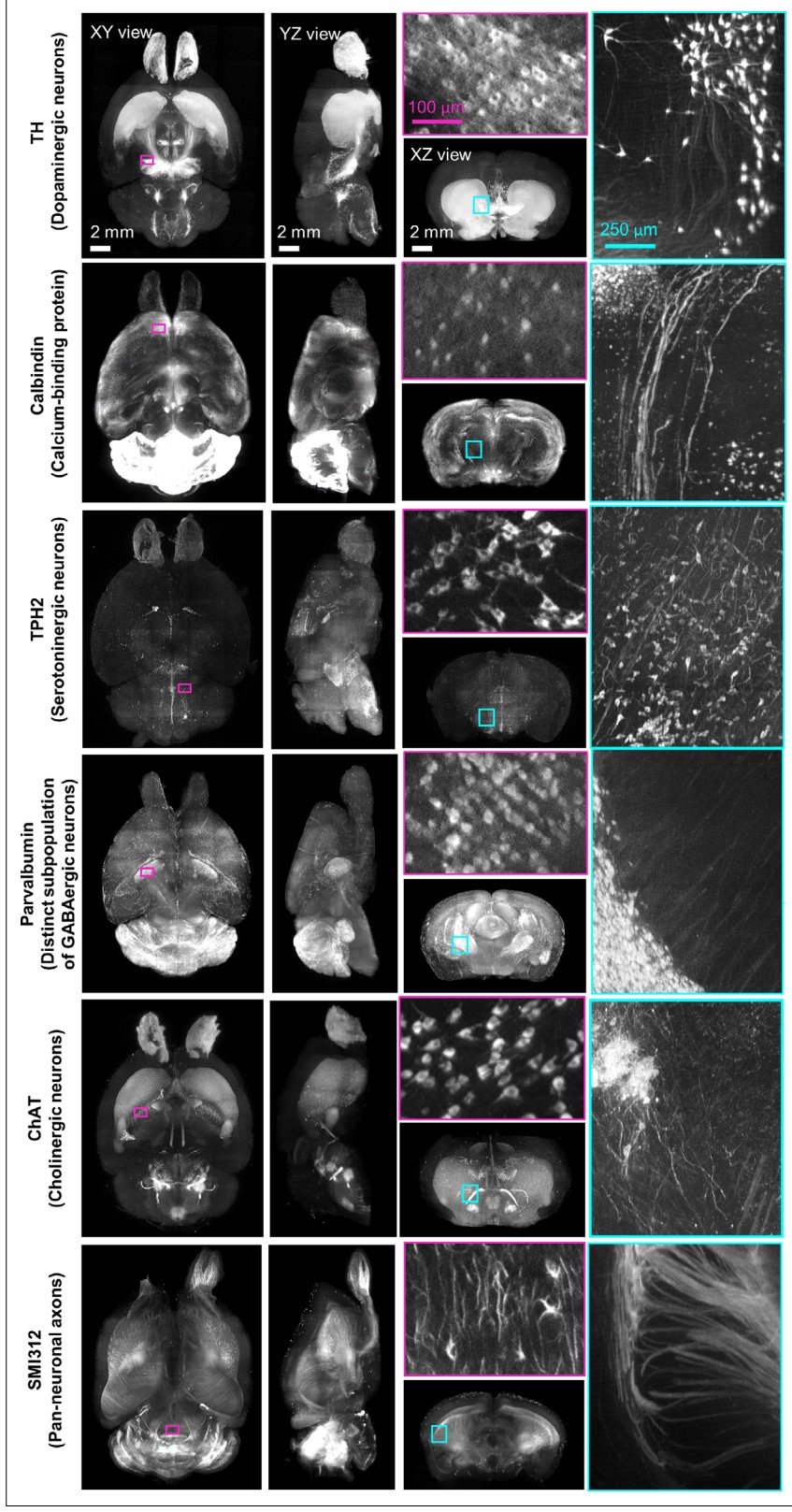

**Figure 4.** Multi-round immunostaining of FFPE-HIF-Clear whole mouse brain (HIF-Clear+). Light-sheet microscopy imaging datasets of six-round immunolabeling performed on one whole FFPE mouse brain. Projection images of the horizontal (XY), sagittal (YZ), and transverse (XZ) views are shown. Projection images (200 μm thick) of magenta-

*Figure 4 continued on next page*

*Figure 4 continued*

lined and cyan-lined regions marked in the XY and XZ views are magnified and displayed within correspondingly colored frames. Scale bars are indicated in the first row. FFPE, formalin-fixed paraffin-embedded.

The online version of this article includes the following video for figure 4:

**Figure 4—video 1.** Spatial distribution of the six biological markers shown in *Figure 4*.
https://elifesciences.org/articles/93212/figures#fig4video1

## HIF-Clear facilitates retrospective 3D analyses of neuronal diseases

FFPE processing and sectioning are standard procedures in pathology departments, and FFPE sections are routinely subjected to targeted chemical staining or immunohistochemistry for morphological observations, biomarker validations, or clinical diagnoses (*Berg et al., 2011*; *Steeghs et al., 2020*). However, it is challenging to obtain brain-wide information from conventional 3–5-µm-thick FFPE sections for studies of neurodegenerative diseases (*Golub et al., 2015*). Moreover, biomarker quantifications and structural observations of 2D sections can be biased by sectioning positions and orientations. By applying HIF-Clear/HIF-Clear+, 3D histological information can be collected from FFPE specimens, thereby providing more accurate biomarker quantifications, brain-wide neuronal circuitry, and cell analyses. Moreover, HIF-Clear/HIF-Clear+ may be performed on archived patient samples hosted in neuronal disease brain banks to establish a 3D biomarker atlas for neurological diseases, maximizing the utility of archived specimens. Such an atlas would represent an invaluable resource for retrospective research on biomarker distributions throughout the entire neural network, enabling tracking of their changes and interactions. The resulting comprehensive imaging datasets could also be deployed to train deep learning models, explore potential therapeutic targets, and predict early diagnostic markers.

## FFPE-HIF-Clear as an archival and delipidation strategy for clearing centimeter-sized tissue samples

The development of tissue clearing has revolutionized neuroscientific research. However, current tissue-clearing protocols impose a 24 hr fixation limit on mouse brains, requiring immediate tissue clearing and labeling upon animal sacrifice. HIF-Clear offers a solution to this predicament by enabling long-term preservation of mouse brains in FFPE blocks. Accordingly, clinical samples can be conveniently transported to imaging facilities and staining decisions can be postponed until essential analytical results, such as RNA sequencing, become available. Additionally, HIF-Clear minimizes excessive use of experimental mice in time-course studies by allowing brain samples in FFPE blocks after sacrifice, adhering to 3Rs principles. Consequently, researchers can decide on whole-brain staining approaches upon completing experiments, such as for toxicological or pharmacological targets, thereby avoiding the need to repeat animal experiments.

Another possible application of FFPE-HIF-Clear is to use it as a tissue-clearing method for removing lipids. Since paraffin embedding, dewaxing, and heat-induced antigen retrieval are commonly used pathological procedures, FFPE-HIF-Clear can be quickly, easily, and efficiently executed in the pathology department of hospitals or research institutions, reinforcing its potential utility in clinical research.

## Microscopy options for imaging centimeter-sized specimens

Optical sectioning techniques are crucial for obtaining high-quality volumetric images. Techniques such as confocal microscopes, multi-photon microscopy, and light-sheet microscopy filter out-of-focus signals, resulting in sharp images of individual planes. In our study, we used light-sheet microscopy and multi-point confocal (i.e., spinning disc) for imaging centimeter-sized specimens because of their scanning speeds. While two-photon and confocal microscopy offer high-resolution imaging of smaller volumes, they are not ideal for scanning entire tissues because of their prolonged scanning times.

Nonoptical sectioning wide-field fluorescence microscopes, like the Olympus BX series or Zeiss Axio imager series, can also be used to scan samples up to about 3.5 mm thick with long working distance objective lenses. In these cases, deconvolution algorithms are required to eliminate out-of-focus signals. However, it should be noted that the epifluorescence system might reduce fluorescent intensity in deeper regions within the samples.

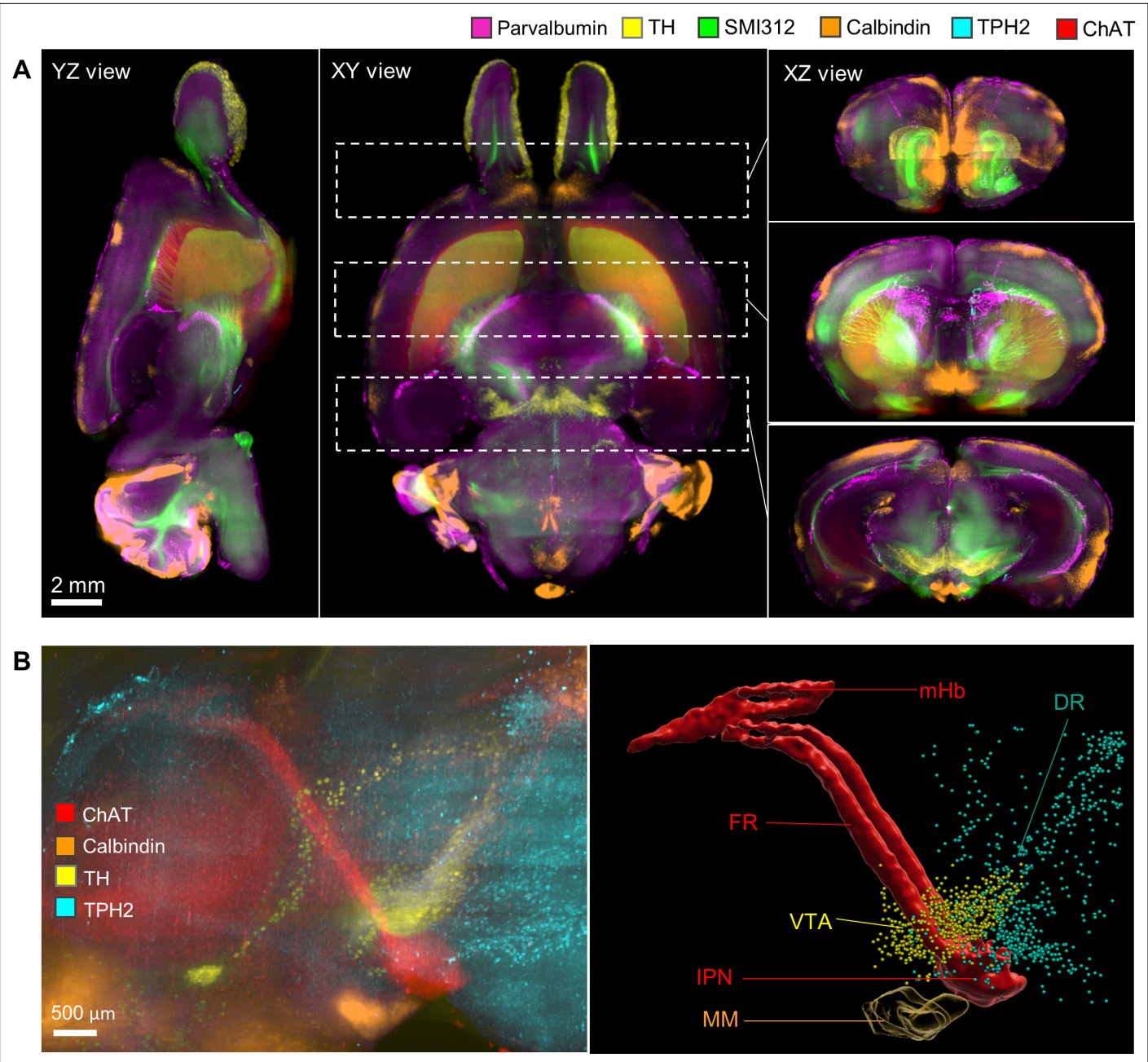

**Figure 5.** Merged multichannel neuron circuitry image generated by HIF-Clear+. (**A**) Merged multichannel image generated using the datasets from *Figure 3*. Optical sections from the horizontal (XY) and sagittal (YZ) perspectives are presented. For the transverse (XZ) views, projection images of the regions encompassed by the dotted lines in the XY view are displayed. (**B**) The spatial relationship of the Hbn-IPN circuitry, dopaminergic (TH), serotoninergic (TPH2), and calbindin+GABAergic neurons. Fluorescence (right) and segmented (left) images are shown. mHb, medial habenula; FR, fasciculus retroflexus; IPN, interpeduncular nucleus; SuM, medial mammillary area of the hypothalamus; DR, dorsal raphe nuclei.

## Heating source and buffer choices for heat-induced antigen retrieval in HIF-Clear procedure

Early efforts to clear and label archived FFPE specimens have only achieved limited labeling depths (*Chen et al., 2019*; *Hughes et al., 2014*; *Lai et al., 2018*; *Nojima et al., 2017*; *Tanaka et al., 2017*; *Tanaka et al., 2018*), likely due to a lack of effective antigen retrieval. Protein denaturants employed in hydrophilic tissue-clearing protocols, such as SDS (in CLARITY, PACT, FLASH) and urea (in CUBIC and Scale), enhance signal intensity and specificity for some types of specimens, such as cryosections

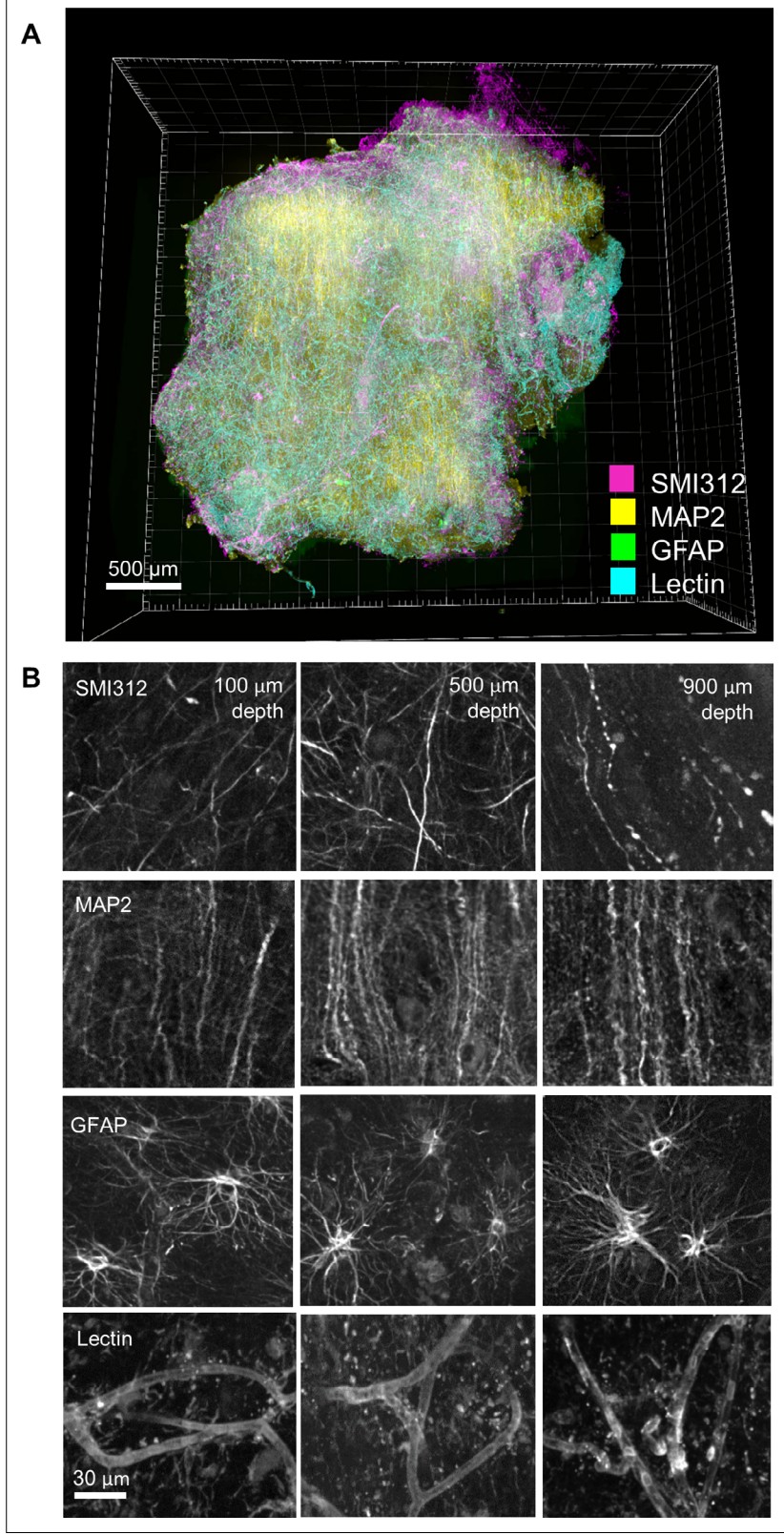

**Figure 6.** Multi-round immunostaining of an FFPE-HIF-Clear human brain specimen. Three-round immunolabeling was performed on an FFPE human brain specimen collected from a patient with cerebral hemorrhage. Images were acquired using multi-point confocal microscopy. (**A**) 3D rendering of the entire specimen. (**B**) Images depicting merged and single channels of an optical section from the selected area (white frame) in (**A**) at depths

*Figure 6 continued on next page*

*Figure 6 continued*

of 100, 500, and 900 μm. GFAP, glial fibrillary acidic protein; SMI312, pan-axonal marker; MAP2, microtubule-associated protein 2 (a dendritic marker); FFPE, formalin-fixed paraffin-embedded.

The online version of this article includes the following figure supplement(s) for figure 6:

**Figure supplement 1.** Comparison of staining qualities of human and mouse brain tissues.

or well-controlled formalin-fixed but non-paraffin-embedded tissues. However, their utility for FFPE tissues is less apparent, with temperature being considered a primary factor in their applicability in such instances.

Heat-induced antigen retrieval has been widely used since first being reported by Shi et al. in 1991, with many protocol variations since being developed, including the use of different heat sources (e.g., water baths, steamers, microwave ovens, autoclaves, and pressure cookers) and buffer choices (low molarity buffers of acid or alkaline pH) (*Krenacs et al., 2010*). Nevertheless, a unifying principle is to heat the fixed tissue sections in buffer solutions at or above 100°C for up to 30 min without boiling (*Taylor et al., 1996*; *Krenacs et al., 2010*). In our study, we used a pressure cooker as the heat source to achieve a temperature of ~121°C, thereby ensuring temperature stability and time efficiency. Any heating condition that is effective for FFPE sections should also work for centimeter-scale FFPE specimens (e.g., using water baths at 95–99°C), but their effects on delipidation require further testing.

The choice of antigen retrieval solutions depends on the antibody being deployed. For example, EDTA solutions are favorable for antibodies against phosphorylated tyrosine (*Krenacs et al., 2010*; *Pileri et al., 1997*). Generally, alkaline pH solutions exhibit higher antigen retrieval efficiency, but they are also more tissue-destructive and result in higher background signal compared to lower pH solutions (*Shi et al., 2001*; *Krenacs et al., 2010*). In our study, we tested the addition of 1% Tween 20 to two common antigen retrieval solutions, citrate buffer (pH 6.0) and Tris-EDTA (pH 9.0), both of which yielded satisfactory staining results (*Figure 8—figure supplement 1*), demonstrating that HIF-Clear facilitates the choice between acidic or alkaline antigen retrieval solutions based on antibody preference without compromising the delipidation effect.

## Staining strategies and epoxy/hydrogel pretreatments in HIF-Clear

In *Figure 1D*, we propose two staining strategies for samples with thicknesses less than 500 μm and greater than 1 mm: passive immunolabeling and active immunolabeling. In passive immunolabeling, antibodies penetrate and reach their targets solely through diffusion, without any additional force. It takes ~2 mo to passively stain a whole mouse brain (*Susaki et al., 2014*; *Tainaka et al., 2018*). Compared to passive immunolabeling, active immunolabeling uses an external force, such as pressure and electrophoresis, to facilitate antibody penetration and therefore significantly speed up the staining process, reducing the required staining time for a whole mouse brain to 1 d. However, the harsh conditions, such as pressure and heat, caused by external forces might damage specimens. To protect specimens from the harsh conditions caused by active staining, specimens could be strengthened by treatment with epoxy or acrylamide monomer to form a tissue-epoxy or tissue-hydrogel hybrid (*Chung et al., 2013*; *Park et al., 2019*). Laboratories that do not have adequate devices or handle small specimens could use passive immunolabeling instead and skip the step of epoxy or hydrogel pretreatment.

## Limitations

We acknowledge that there are some limitations to HIF-Clear application, which we wish to address. Firstly, the FFPE tissues we tested in this study have been limited to mouse and human brain tissues fixed under well-controlled conditions. Differences in tissue fixation procedures at different hospitals and tissue banks may affect the clarity and immunostaining efficiency of dewaxed FFPE tissues. In the demonstration of HIF-Clear to 3-month fixed specimens, we observed that pontine reticular nucleus (*Figure 3A*, yellow arrowheads) lose TH-positive signals after long-term fixation. The fluorescence intensity was more affected by fixation with formalin, which is methanol-stabilized and stronger, than with PFA. The results indicate that a stronger antigen retrieval method may be a possible solution. However, achieving the right balance between antigen retrieval efficiency and tissue integrity will require additional testing and investigation. Secondly, HIF-Clear has not yet been extensively tested

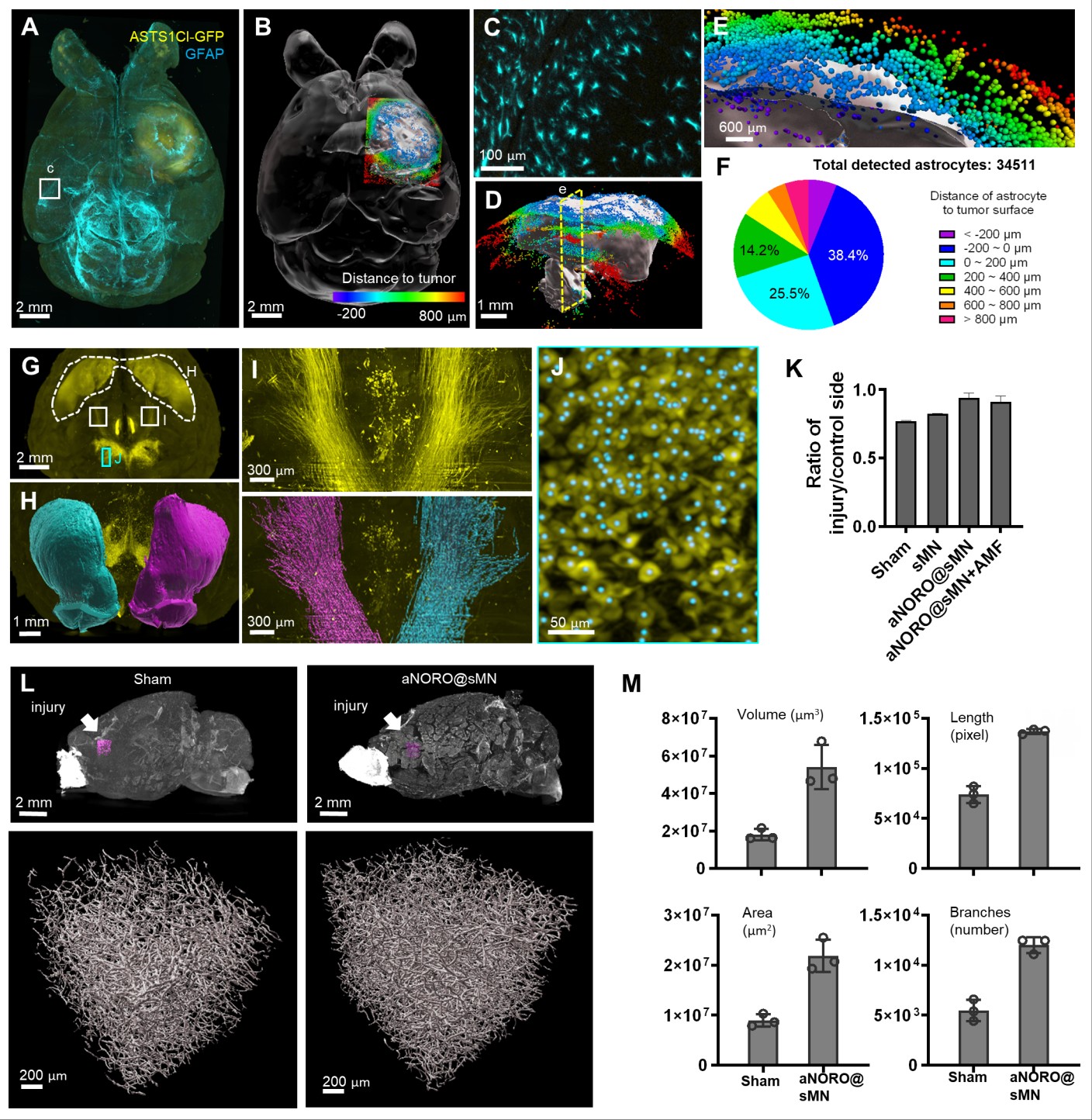

**Figure 7.** Applications of HIF-Clear in disease models. (**A–F**) HIF-Clear reveals cell–tumor relationships in an astrocytoma model. (**A**) Projection light-sheet image of an FFPE-HIF-Clear mouse brain with a GFP-expressing astrocytoma xenograft (ALTS1C1 cells, see 'Materials and methods') stained for GFP (yellow) and GFAP (cyan). (**B**) Segmented tumor (white) and surrounding astrocytes (colored). The astrocytes are colored according to the color-coded scale of cell-to-tumor distance. (**C**) Magnification of the region marked in (**A**), showing astrocyte morphology. (**D**) Sagittal view of the segmented tumor and color-coded astrocytes. (**E**) Magnification of the clipping plane marked in (**D**, yellow dashed line) showing astrocytes inside (dark blue, purple) and outside (red, orange, yellow, green, cyan, blue) the tumor. (**F**) Quantification of detected astrocytes and their classification according to distance to the tumor surface. (**G–M**) HIF-Clear reveals brain damage and the therapeutic effect of alternating magnetic field-responsive NO-release octahedrons (aNORO) administered using a paracetamol-coated silk-based microneedle (sMN) in a traumatic brain injury (TBI) mouse model. (**G–J**) Segmentation of dopaminergic regions in FFPE-HIF-Clear TBI mouse brains. (**G**) Whole-brain projection light-sheet image focusing on tyrosine hydroxylase (TH)-positive

*Figure 7 continued on next page*

*Figure 7 continued*

regions. (**H**) Frontal view of the segmented striatum (dashed white line in [**G**]). Magenta: the injured side; cyan: the contralateral side. (**I**) Horizontal views of the nigrostriatal fiber tract (paired white boxes in [**G**]); fluorescence (top) and segmented (bottom) images are shown. Magenta: the injured side; cyan: the contralateral side. (**J**) Detection of dopaminergic cells in the substantia nigra (SN). Magnification of the marked area in (**G**), including the TH fluorescence signal (yellow) and segmented cells (cyan dots). (**K**) Comparison of the therapeutic effects of different treatments. n = 3 (three indicators: striatum volume, nigrostriatal fiber tract volume, SN cell number), mean ± SD, *p<0.05, **p<0.01, ***p<0.001, ****p<0.0001, one-way ANOVA with Tukey's multiple comparison test. (**L**) Reconstruction of blood vessels in TBI brains with or without treatments. The regions of interest (ROIs) selected to assess angiogenesis are indicated in the top row (magenta boxes, directly below the injury site). 3D reconstructions of blood vessels are shown in the bottom row. (**M**) Quantification of blood vessel volume, blood vessel surface area, number of blood vessel branches, and blood vessel length. n = 3, mean ± SD have been plotted. FFPE, formalin-fixed paraffin-embedded.

The online version of this article includes the following video and figure supplement(s) for figure 7:

**Figure supplement 1.** Segmentation of dopaminergic regions and blood vessels in PFA-SDS traumatic brain injury (TBI) mouse brains.

**Figure supplement 2.** Quantifications of FFPE-HIF-Clear and PFA-SDS samples result in identical statistical trends.

**Figure 7—video 1.** Spatial relationships of the tumor and surrounding astrocytes in *Figure 7A–E*.
https://elifesciences.org/articles/93212/figures#fig7video1

**Figure 7—video 2.** 3D visualization of the segmented blood vessels shown in *Figure 7L*.
https://elifesciences.org/articles/93212/figures#fig7video2

on tissues other than the brain. Thus, its applicability and effectiveness for other tissues remain to be established. Third, the size of the samples used in this study is mostly in the range of 2–3 cm$^3$. Larger samples may require a prolonged incubation time and/or increased Tween 20 concentration. Fourth, HIF-Clear is not compatible with endogenous fluorescence due to a reduction in fluorescence intensity caused by xylene and alcohol used in paraffin processing. Researchers who need to directly observe genetically encoded fluorescent proteins can utilize tissue-clearing methods such as 3DISCO, X-CLARITY, and CUBIC, which have been shown to minimize the decrease in fluorescence intensity. On the other hand, if researchers need to visualize transgenic fluorescent proteins along with other biomarkers, they can use HIF-Clear for delipidation and boost-immunolabeling to visualize the transgenic fluorescent proteins. Fifth, HIF-Clear+ may not be suitable for cases necessitating iterative staining with the same antibody. In addition, the applicability of HIF-Clear+ to antibodies not tested in the study requires further testing and validation. Finally, whether HIF-Clear-treated tissues retain sufficient molecular features and structural integrity for omics analysis warrants verification. Further research and testing will help to address these limitations and improve the applicability and effectiveness of the HIF-Clear pipeline we present herein.

## Conclusions

In conclusion, as a method enabling multi-round immunolabeling of centimeter-scale archived FFPE specimens, HIF-Clear/HIF-Clear+ presents new possibilities for researching neural disease and could serve as a powerful tool for neuronal circuitry analyses and pathology. Its integration with conventional pathological procedures also renders it readily applicable in clinical settings. Through further validation across various organs and testing of its compatibility with spatial omics techniques, HIF-Clear could potentially catalyze broader and more comprehensive investigations in both the fundamental and clinical research domains.

## Materials and methods

### Experimental animals and human samples

We used 8-week-old CD-1 male mice to develop the HIF-Clear pipeline. All animal procedures and handling complied with guidelines from the Institutional Animal Care and Use Committee of Academia Sinica (IACUC protocol no.: 12-05-370), Taiwan. Information on the astrocytoma and TBI mouse models is provided in the respective subsections below. The analyses involving human participants were reviewed and approved by the Institutional Review Board of National Taiwan University Hospital (IRB no.: 202203079RINC), and all participants provided signed informed consent.

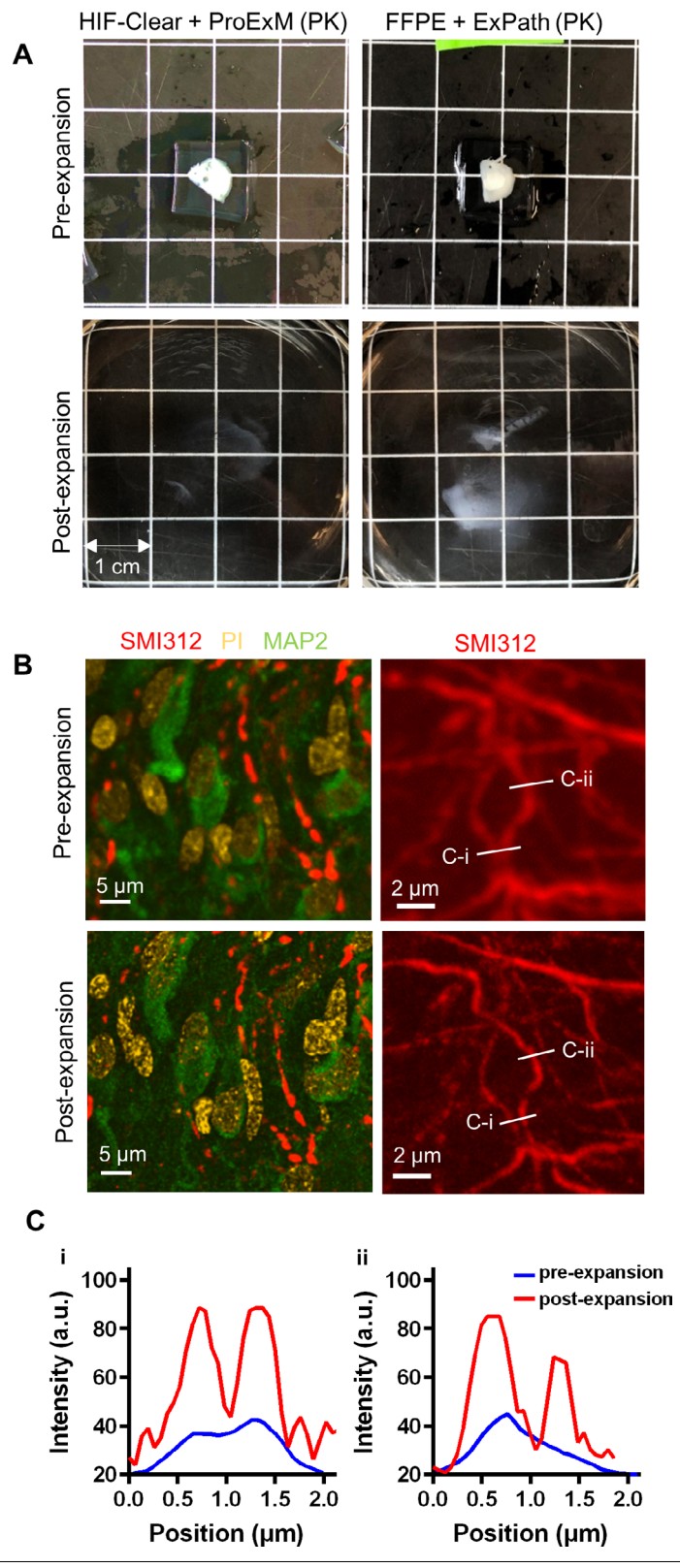

**Figure 8.** Expansion microscopy on an FFPE-HIF-Clear human brain specimen. (**A**) Gross views of expansion of a 1 mm FFPE specimen. (**B**) Fluorescence images of an FFPE human brain pre- (top row) and post-expansion (bottom row). Left: merged images of multiplexed staining; right: fluorescence images of SMI312 staining. (**C**) Profiles of

*Figure 8 continued on next page*

*Figure 8 continued*

axon signal intensity taken along the white lines shown in the images of the right panel in (**B**). The scale bar has been divided by the expansion factor. FFPE, formalin-fixed paraffin-embedded .

The online version of this article includes the following figure supplement(s) for figure 8:

**Figure supplement 1.** Effects of various antigen retrieval conditions on formalin-fixed paraffin-embedded (FFPE) mouse brains.

## Astrocytoma animal model

All animal procedures and handling complied with guidelines from the Institutional Animal Care and Use Committee of National Tsing Hua University (IACUC protocol no.: 109067), Taiwan. ALTS1C1-GFP cells ($1 \times 10^5$ cells) were implanted into the brains of 8-week-old C57BL/6 mice according to a previously published procedure (*Wang et al., 2012*). In brief, $1 \times 10^5$ ALTS1C1-GFP cells were injected into the C57BL/6 mice under anesthesia at a depth of 2.5 mm and 2 mm lateral to the midline and 1 mm posterior to the bregma. After injection, bone wax (ETHICON, W810, Somerville, NJ) was applied to seal the drilled hole. Eighteen days after tumor implantation, mice were sacrificed after cardiac perfusion.

## TBI mouse model

All animal procedures and handling complied with guidelines from the Institutional Animal Care and Use Committee of National Tsing Hua University (IACUC protocol number: 110081). We divided 7-week-old C57BL/6 female mice into four groups: (1) untreated (sham), (2) treated with a paracetamol-coated silk-based microneedle (sMN), (3) treated with alternating magnetic field-responsive NO-release octahedrons (aNORO@sMN), and (4) treated with aNORO@sMN and an alternating magnetic field (aNORO@sMN+AMF). To perform TBI, we used an electric drill to create a hole in the skull at the left motor cortex (M1 and M2) and used a 2-mm-diameter punch to cause a 1.5-mm-deep injury. For all treatments except sham, the microneedle was implanted 1 d post-injury to avoid excessive swelling in the injured area. For the aNORO@sMN and aNORO@sMN+AMF groups, the NO-release octahedrons (aNORO) were embedded in a silk microneedle (sMN). For the AMF-treated group, a magnetic field at a power of 3.2 kW and a frequency of 1 MHz was applied for 5 min per day until the mice were sacrificed. Additional experimental details are as described in *Chan et al., 2023* Mice were sacrificed 45 d after TBI induction.

## Sample collection and preparation

For mouse brain samples, the mice were euthanized by injecting them with 0.2 mL of a 30% urethane saline solution. The euthanized mice were perfused with 20 mL of cold phosphate-buffered saline (PBS) and 20 mL of 4% PFA. The collected brains were further fixed using 4% PFA at 4°C for 24 hr. Human brain tissue specimens were surgically collected from the temporal lobe of patients and fixed with 4% PFA at 4°C for 24 hr. For paraffin embedding, fixed specimens were dehydrated and processed in a tissue processor (Tissue-Tek VIP 5 Jr., Sakura Finetek Japan Co. Ltd., Japan) under vacuum conditions and then embedded in paraffin wax.

## SHIELD processing, SDS-electrophoretic delipidation, and FLASH delipidation

PFA-fixed specimens were incubated in SHIELD-OFF solution at 4°C for 96 hr, followed by incubation for 24 hr in SHIELD-ON solution at 37°C. All reagents were prepared using SHIELD kits (LifeCanvas Technologies, Seoul, South Korea) according to the manufacturer's instructions. For SDS-electrophoretic delipidation, SHIELD-processed specimens were placed in a stochastic electro-transport machine (SmartClear Pro II, LifeCanvas Technologies) running at a constant current of 1.2 A for 5–7 d. For FLASH delipidation, the SHIELD-processed specimens were placed in FLASH reagent (4% w/v SDS, 200 mM borate) and then incubated at 54°C for 18 hr (*Messal et al., 2021*). The delipidated specimens were washed with PBST at room temperature for at least 1 d.

### Evaluating the extent of delipidation

Serial 20-µm-thick cryosections were cut from mouse brain slices (2 mm thick) of various treatment conditions for subsequent DiD or Oil red O staining. For DiD staining, cryosections (that were of ~0–40 µm depth) were post-fixed with 4% PFA at room temperature for 5 min. The sections were then rinsed with distilled water and immersed in PBST (PBS with 0.1% Triton X-100) for 10 min to remove OCT. These sections were stained with 0.0025 µM DiD/PBST solution at room temperature for 1 hr, before rinsing them with distilled water and washing with 1% SDS for several seconds to remove excess DiD. The slides were mounted in 50% glycerol and imaged under a Zeiss 780 confocal microscope using a 10×0.75 NA Plan-Apochromat objective (Zeiss). For Oil red O staining, the sections (that were of ~40–80 µm depth) were post-fixed with a 4% formal-calcium solution for 5 min and then rinsed with distilled water. The fixed sections were pre-incubated in 60% isopropanol for 5 min and then stained with fresh Oil red O working solution (0.3% Oil red O in 60% isopropanol) for 15 min. The sections were briefly washed with 60% isopropanol and mounted in glycerine jelly. Images were acquired using a 3DHISTECH Pannoramic 250 slide scanner with a 40×0.95 NA Plan-Apochromat objective (Zeiss). Details of statistical analyses are described in the section 'Quantification and statistics'.

### Comparison of brain region volume

Autofluorescence-channel imaging datasets of whole mouse brains were aligned to the BrainGlobe Atlas API (*Claudi et al., 2020*) using the aMAP tool (*Niedworok et al., 2016*). The volume of each brain subregion was then obtained and organized based on anatomical hierarchy using the Allen Brain Atlas. The 679 subregions were merged into 78 brain regions at the same anatomical hierarchy as the hippocampus. The brain region volume of right and left hemisphere of the FFPE-HIF-Clear mouse brain was compared and plotted using GraphPad Prism 9.5.1 (GraphPad Software Inc, San Diego, CA).

### HIF-Clear pipeline

The HIF-Clear pipeline is illustrated in *Figure 1D* and described in detail below.

### Deparaffination and rehydration

FFPE blocks were incubated at 65–70°C to melt the paraffin. Residual paraffin was removed through immersion in xylene for 24 hr, with the xylene being changed at least twice. Dewaxed brains were rehydrated with 100, 100, 95, 85, 75, and 55% alcohol diluted with distilled water. The rehydrated brains were then washed with PBS.

### Epoxy processing

Specimens were processed with commercialized SHIELD kit as described above.

### Optimized antigen retrieval

Specimens were incubated overnight at 37°C in a modified citrate buffer (10 mM sodium citrate, 1% Tween 20, pH 6.0). The specimens and buffer were then heated for 10 min at 121°C in a pressure cooker (Bio SB, Santa Barbara, CA) and kept in the pressure cooker until the temperature dropped below 100°C (for ~15 min). Next, the specimens were removed from the cooker and cooled for 10 min at room temperature. The specimens were then washed three times in PBST, with each wash lasting 1 hr.

### Electrophoretic immunolabeling (active staining)

The procedure was modified from the previously published eFLASH protocol (*Yun et al., 2019*) and was conducted in a SmartLabel System (LifeCanvas Technologies). The specimens were preincubated overnight at room temperature in sample buffer (240 mM Tris, 160 mM CAPS, 20% w/v D-sorbitol, 0.9% w/v sodium deoxycholate). Each preincubated specimen was placed in a sample cup (provided by the manufacturer with the SmartLabel System) containing primary, corresponding secondary antibodies and lectin diluted in 8 mL of sample buffer. Information on antibodies, lectin, and their optimized quantities is detailed in *Supplementary file 1*. The specimens in the sample cup and 500 mL of labeling buffer (240 mM Tris, 160 mM CAPS, 20% w/v D-sorbitol, 0.2% w/v sodium deoxycholate) were loaded into the SmartLabel System. The device was operated at a constant voltage of 90 V

with a current limit of 400 mA. After 18 hr of electrophoresis, 300 mL of booster solution (20% w/v D-sorbitol, 60 mM boric acid) was added, and electrophoresis continued for 4 hr. During the labeling, the temperature inside the device was kept at 25°C. Labeled specimens were washed twice (3 hr per wash) with PTwH (1×PBS with 0.2% w/v Tween-20 and 10 µg/mL heparin) (*Renier et al., 2014*), and then post-fixed with 4% PFA at room temperature for 1 d. Post-fixed specimens were washed twice (3 hr per wash) with PBST to remove any residual PFA.

## RI matching

Before imaging, the specimens were RI-matched by being immersed in NFC1 (RI = 1.47) and NFC2 (RI = 1.52) solutions (Nebulum, Taipei, Taiwan). Each immersion lasted for 1 d at room temperature. Alternatively, RI matching can also be accomplished by immersing specimens in a 1:1 dilution of CUBIC-R (*Tainaka et al., 2018*) for 1 d, followed by pure CUBIC-R for an additional day.

## Volumetric imaging and 3D visualization

For centimeter-scale specimens, images were acquired using a light-sheet microscope (SmartSPIM, LifeCanvas Technologies) with a 3.6× customized immersion objective (NA = 0.2, working distance = 1.2 cm). For samples <3 mm thick, imaging was performed using a multipoint confocal microscope (Andor Dragonfly 200, Oxford Instruments, UK) with objectives that were UMPLFLN10XW (10×, NA = 0.3, working distance = 3.5 mm), UMPLFLN20XW (20×, NA = 0.5, working distance = 3.5 mm), and UMPLFLN40XW (40×, NA = 0.8, working distance = 3.3 mm). 3D visualization was performed using Imaris software (Imaris 9.5.0, Bitplane, Belfast, UK).

## Photobleaching

Immunolabeled and imaged specimens were placed in a multi-well plate and immersed in RI-matching solutions so that they retained transparency. The plate was then sealed with paraffin. A 100 W projection lamp with an LED array was placed on the plate to quench fluorescence signals. A representative photobleaching apparatus constructed using off-the-shelf components is shown in *Figure 3—figure supplement 4a*. In our experience, ~18 hr photobleaching is sufficient for a 2-mm-thick sample, whereas 3 d is required for a whole mouse brain (approximately 8 mm to 1 cm in thickness).

## Raman microscopy

A Yb:KGW laser (Carbide, Light Conversion) emitting a 190 fs, 200 kHz, 20 W pulse train at a 1030 nm wavelength was used to generate a supercontinuum via a double-pass multiple-plate continuum module. The resulting spectrum (600–1300 nm) was spectrally sliced by tunable color filters (3G LVLWP, LVSWP, and LVFBP, Delta) to provide pump (725 nm) and Stokes (914 nm) beams, which could address the Raman shift (2850 cm$^{-1}$) of lipids. The two beams were temporally and spatially overlapped, and then guided into a commercial upright microscope (Axio Examiner.Z1, Zeiss) with a scanning unit (LSM7MP, Zeiss) to achieve raster scanning on the x–y plane. A single ×20 water immersion objective (UMPLFLN 20XW, Olympus) was used to focus the combined laser beams on the mouse sample. Epi-CARS signal at 601 nm was spectrally separated from the incident radiation with a bandpass filter (BP 565-610, Zeiss) and detected using a photomultiplier tube.

## Multichannel image registration

For six-round immunolabeling of a single whole mouse brain, each dataset included one structural reference channel stained with lectin and one antibody-stained channel. The first round of staining served as the standard brain, with lectin channel images of each subsequent dataset being registered to the lectin channel of the standard brain using the Elastix toolbox (*Klein et al., 2010*). The transformation parameters obtained from rigid and B-spline deformable registration of the lectin channels were then applied to the antibody-stained channels. The transformed images were merged and visualized using Imaris software. In the case of three-round immunolabeling of human specimens, the image datasets from the three rounds were registered and resampled using Amira software (Thermo Fisher Scientific, Waltham, MA), and then merged and visualized using Imaris software.

## Expansion microscopy and ExPath

Expansion microscopy was performed using the proExM protocol (*Jenkins et al., 2018*). A 1-mm-thick HIF-Clear-processed FFPE mouse brain slice was incubated in anchoring solution (0.01% w/v methacrylic

acid N-hydroxysuccinimide ester in 1× PBS) at 4°C for 24 hr. After anchoring, the sample was washed twice (5 min each time) with PBS at room temperature and then rinsed twice (5 min each time) in acrylamide monomer solution (2 M NaCl, 8.625% w/w sodium acrylate, 2.5% w/w acrylamide, 0.15% w/w N,N'-methylene-bisacrylamide in 1× PBS) at 4°C. For gelation, the specimen was pre-incubated in gelation solution (0.02% w/w ammonium persulfate, 0.02% w/w tetramethyl-ethylenediamine, and 0.01% w/w 4-hydroxy-2,2,6,6-tetramethylpiperidin-1-oxyl in monomer solution) for 5 min at 4°C, and then moved into fresh gelation solution for an additional 25 min. The specimen in gelation solution was then transferred to a humidified 37°C incubator for 2 hr. The gel was then immersed in a digestion solution (8 units/mL proteinase K, 50 mM Tris, 1 mM EDTA, 0.5% w/v Triton X-100, 1 M NaCl, pH 8.0) at 37°C for 4 hr. After digestion, the gel was immersed in an excess of deionized water for 0.5–2 hr to expand. This step was repeated 3–5 times with fresh water until the size of the sample ceased to increase.

For the ExPath experiment, a 1-mm-thick FFPE mouse slice was dewaxed, rehydrated, and underwent the same anchoring and gelation steps as described above. The gel was digested in ExPath digestion solution (8 units/mL proteinase K, 50 mM Tris, 25 mM EDTA, 0.5% w/v Triton X-100, 0.8 M NaCl, pH 8.0) at 60°C for 3 hr (*Chen et al., 2015*).

## Post-expansion immunolabeling

Digested gels were washed twice (20 min each time) in PBS, and then incubated in a primary antibody cocktail containing anti-SMI312 (BioLegend, 837904, 1:500 dilution), anti-MAP2 (Cell Signaling Technology, #8707, 1:500 dilution), 2% v/v normal goat serum (Jackson ImmunoResearch, 005-000-121, 1:500 dilution), and 0.1% w/v Triton X-100 in 1× PBS at 4°C for 24 hr. The gels were then washed three times (20 min each time) in wash buffer (30 mM sodium chloride, 0.1% w/v Triton X-100 in 1× PBS) and incubated in a secondary antibody cocktail containing Alexa Fluor 647 AffiniPure goat anti-mouse IgG (Jackson ImmunoResearch, 115-605-003, 1:500 dilution), Rhodamine Red-X (RRX) AffiniPure goat anti-rabbit IgG (Jackson ImmunoResearch, 111-295-003, 1:500 dilution), 0.2% v/v normal goat serum, and 0.1% w/v Triton X-100 in 1× PBS at 4°C for 24 hr. The antibody-labeled gels were washed three times (20 min each time) in wash buffer and then placed in deionized water for expansion. For nuclei staining, propidium iodine was added to deionized water at a final concentration of 1 μg/mL during expansion.

## Quantification and statistics

To quantify the extent of delipidation (*Figure 1A*), DiD-stained sections were imaged at an excitation wavelength of 642 nm. To quantify the fluorescence intensity, 10 ROIs of 0.8 × 0.8 mm were selected randomly within the cortex of each specimen. The gray (pixel) value of each ROI was measured using Fiji (*Schindelin et al., 2012*), Mean values and standard deviations (SDs) were plotted using GraphPad Prism 9.5.1 (GraphPad Software Inc). The significance of the difference in mean values was determined by means of one-way ANOVA with Tukey's multiple comparison tests at an α level = 0.05 (*p<0.05; **p<0.01; ***p<0.001; ****p<0.0001).

For the Raman signal quantification shown in *Figure 1F*, gray values of 18 Raman images (0.4 × 0.4 mm) at a different depth for each group were measured using Fiji. The mean values and SD were determined, and the significance of differences was also calculated by means of one-way ANOVA using GraphPad Prism as described above.

For the astrocyte-tumor distance analysis shown in *Figure 7F*, the astrocytes were segmented using the SPOT module of Imaris based on a fluorescence intensity threshold and point spread function size definition. GFP-expressing tumors were segmented using the Surface module of Imaris. Distances of all segments were calculated in Imaris.

To quantify dopaminergic regions in TBI mouse brains, the volume of the striatum and nigrostriatal fiber tracts was calculated using the Surface module of Imaris, and cell number in the substantia nigra was determined using the SPOT module. To demonstrate the similarity of quantification results from FFPE-HIF-Clear and PFA-SDS brains, the quantification data for the injured side of each brain was normalized according to the results for the corresponding contralateral side. Mean values and SD were plotted using GraphPad Prism (N = 3 in *Figure 7K and M* and *Figure 7—figure supplement 2*). The significance of the difference in mean values was determined by means of one-way ANOVA with Tukey's multiple comparison tests at an α level = 0.05 (*p<0.05; **p<0.01; ***p<0.001; ****p<0.0001).

To evaluate angiogenesis, we selected three ROIs (each of 0.6 × 0.6 × 0.6 mm) close to the injury site. The Surface module of Imaris was used to segment the blood vessels and calculate their total volume and surface area. Their length and numbers of bifurcation points were calculated using Vessap (*Todorov et al., 2020*). Erosion and dilation were performed to remove false-negative pixels and avoid false centerline detections. Next, the centerlines were extracted by means of a 3D thinning algorithm (*Lee et al., 1994*). The bifurcation points were detected using the surrounding pixels of each point to define a point that splits into two or more vessels. Mean values, SD, and the significance of differences were calculated by independent *t*-test using GraphPad Prism.

## Acknowledgements

We thank the Brain Research Center, National Tsing-Hua University, Taiwan, and Pathology Core, Institute of Biomedical Sciences, Academia Sinica, Taiwan, for the technical assistance. This work was funded by the grants from National Science and Technology Council (NSTC-111-2636-B-007-007, NSTC-07-2314-B-007-003-MY3, NSTC-111-2321-B-002-016, MOST-110-2321-B-010-006). This work was also supported by the Brain Research Center under the Higher Education Sprout Project, co-funded by the Ministry of Education and the National Science and Technology Council in Taiwan.

## Additional information

### Funding

| Funder | Grant reference number | Author |
| --- | --- | --- |
| National Science and Technology Council | NSTC 111-2636-B-007-007 | Li-An Chu |
| National Science and Technology Council | NSTC 07-2314-B-007-003-MY3 | Chi-Shiun Chiang |
| National Science and Technology Council | NSTC 110-2321-B-010-006 | Hwai-Jong Cheng |
| National Science and Technology Council | MOST 110-2321-B-010-006 | Li-An Chu |
| National Science and Technology Council | NSTC-111–2321-B-002–016 | Li-An Chu |

The funders had no role in study design, data collection and interpretation, or the decision to submit the work for publication.

### Author contributions

Ya-Hui Lin, Conceptualization, Data curation, Formal analysis, Investigation, Visualization, Methodology, Writing - original draft; Li-Wen Wang, Software, Formal analysis, Validation, Visualization, Methodology; Yen-Hui Chen, Resources, Investigation, Methodology; Yi-Chieh Chan, Sheng-Yan Wu, Guan-Jie Huang, Investigation, Methodology; Shang-Hsiu Hu, Shang-Da Yang, Shi-Wei Chu, Kuo-Chuan Wang, Chin-Hsien Lin, Pei-Hsin Huang, Resources; Chi-Shiun Chiang, Hwai-Jong Cheng, Resources, Funding acquisition; Bi-Chang Chen, Conceptualization, Supervision, Project administration, Writing - review and editing; Li-An Chu, Conceptualization, Resources, Supervision, Funding acquisition, Project administration, Writing - review and editing

### Author ORCIDs

Ya-Hui Lin http://orcid.org/0000-0001-9130-5149
Chi-Shiun Chiang http://orcid.org/0000-0002-2581-4129
Shi-Wei Chu http://orcid.org/0000-0001-7728-4329
Li-An Chu http://orcid.org/0000-0002-4092-3024

## Ethics

The analyses involving human participants were reviewed and approved by the Institutional Review Board of National Taiwan University Hospital (IRB No.: 202203079RINC), and all participants provided signed informed consent.

All animal procedures and handling complied with guidelines from the Institutional Animal Care and Use Committee of Academia Sinica (IACUC protocol No.: 12-05-370), Taiwan. For the astrocytoma animal model and TBI animal model, all animal procedures and handling complied with guidelines from the Institutional Animal Care and Use Committee of National Tsing Hua University (IACUC protocol No.: 109067 and 110081, respectively).

Reviewer #1 (Public review): https://doi.org/10.7554/eLife.93212.4.sa1
Reviewer #2 (Public review): https://doi.org/10.7554/eLife.93212.4.sa2
Author response https://doi.org/10.7554/eLife.93212.4.sa3

---

## Additional files

### Supplementary files

• Supplementary file 1. The antibodies used in the study.

• Supplementary file 2. Abbreviations of brain regions mentioned in *Figure 2B*.

• Supplementary file 3. The antibody conditions in the multi-round immunolabeling shown in *Figure 3*.

• MDAR checklist

### Data availability

All images supporting the findings of this study are included within the main figures and supplementary information. The isotropic 2-times downsampled image dataset from the first round of immunolabeling using tyrosine hydroxylase antibodies, as demonstrated in Figure 4 of the HIF-Clear plus protocol, is available on Dryad. The source data for Figures 1A, 1F, 2B, 7F, 7K, 8C, and the supplementary materials to Figure 7 are also accessible on Dryad. Due to the considerable size of the dataset (approximately 12 TB), the original image datasets were not uploaded; however, they are available for research purposes upon request. Please contact Dr. Li-An Chu at lachu@mx.nthu.edu.tw for access. For atlas registration, we employed the AMAP toolbox, as described by Niedworok et al. For further details, refer to the 'Comparison of brain region volumes' subsection in the Materials and methods section and *Niedworok et al., 2016*. For multichannel image merging, the Elastix toolbox, as described by Klein et al., was used. Additional information can be found in the 'Multichannel image registration' subsection of the Materials and methods section and *Klein et al., 2010*.

The following dataset was generated:

| Author(s) | Year | Dataset title | Dataset URL | Database and Identifier |
|---|---|---|---|---|
| Lin Y, Wang LW, Chen YH, Chan YC, Hu SH, Wu SY, Chiang CS, Huang GJ, Yang SD, Chu SW, Wang KC, Lin CH, Huang PH, Cheng HJ, Chen BC, Chu LA | 2024 | Image dataset of HIF-Clear-processed mouse brain | https://doi.org/10.5061/dryad.prr4xgxv7 | Dryad Digital Repository, 10.5061/dryad.prr4xgxv7 |

---

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
