## [Editor Report · eLife assessment]

The reprocessing and reanalysis of archived samples can yield further insights from past experiments. Here, a **useful** procedure to perform tissue clearing and immunolabeling on large-scale formalin-fixed paraffin-embedded brain specimens is **convincingly** evaluated on a set of archival pathology specimens, and its applicability to further such samples is analyzed. This method will be of interest to both neuroscientists and pathologists.

---

## [Referee Report · Reviewer #1 (Public review)]

In this study, Lin et al developed a protocol termed HIF-Clear, to perform tissue clearing and labelling on large-scale FFPE mouse brain specimens. They have optimized protocols for dewaxing and adequate delipidation of FFPE tissues to enable deep immunolabelling, even for whole mouse brains. This was useful for the study of disease models such as in an astrocytoma model to evaluate spatial architecture of the tumour and its surrounding microenvironment. It was also used in a traumatic brain injury model to quantify changes in vasculature density and differences in monoaminergic innervation. They have also demonstrated the potential of multi-round immunolabelling using photobleaching, as well as expansion microscopy with FFPE samples using Hif Clear.

Comments on revised version:

The revised manuscript by Lin et al is much improved with a more detailed methods description. There are only a few minor comments for the authors that are still valid:

- Some procedures, including the basic HIF-Clear protocol, seem to produce marked tissue expansion that is not mentioned in the manuscript. Users should take this fact into consideration when making measurements.

- The authors have provided a comparison between mouse and human brain samples in Figure S12. However, it is misleading to mention that the "fluorescent signals are comparable at varying depth" as the figure clearly showed a lack of continuous staining especially for SMI312 at 900um depth, and human brain tissue showed considerably increased background signal (likely due to endogenous lipofuscin which has autofluorescent properties). Also, This is difficult to assess in the present design of the experiment because, at different depths, the tissue and the antigen may change themselves... making it difficult to make a direct staining comparison with other depths.

---

## [Referee Report · Reviewer #2 (Public review)]

The manuscript details an investigation aimed at developing a protocol to render centimeter-scale formalin-fixed paraffin-embedded specimens optically transparent and suitable for deep immunolabeling. The authors evaluate various detergents and conditions for epitope retrieval such as acidic or basic buffers combined with high temperatures in entire mouse brains that had been paraffin-embedded for months. They use various protein targets to test active immunolabeling and light-sheet microscopy registration of such preparations to validate their protocol. The final procedure, called MOCAT pipeline, briefly involves 1% Tween 20 in citrate buffer, heated in a pressure cooker at 121 {degree sign}C for 10 minutes. The authors also note that part of the delipidation is achieved by the regular procedure.

Major Strengths

- The simplicity and ease of implementation of the proposed procedure using common laboratory reagents distinguish it favorably from more complex methods.

- Direct comparisons with existing protocols and exploration of alternative conditions enhance the robustness and practicality of the methodology.

Final considerations

The evidence presented supports the effectiveness of the proposed method in rendering thick FFPE samples transparent and facilitating repeated rounds of immunolabeling.

The developed procedure holds promise for advancing tissue and 3D-specific determination of proteins of interest in various settings, including hospitals, basic research, and clinical labs, particularly benefiting neuroscience research.

The methodological findings suggest that MOCAT could have broader applications beyond FFPE samples, differentiating it from other tissue-clearing approaches in that the equipment and chemicals needed are broadly accessible.

---

## [Author Response]

The following is the authors’ response to the previous reviews.

**Reviewer #2 (Public Review):**
Major Weaknesses:The assertion that MOCAT can be rapidly applied in hospital pathology departments seems overstated due to the limited availability of light-sheet microscopes outside research labs. In the first rebuttal letter, authors explain the limitations of other microscopes more readily available in hospitals. This explanation relies on your own investigations and practical experience on the matter, so including them in some part of the manuscript would be beneficial.

We appreciate the reviewer's comments and have added a discussion on the limitations of microscopes that are more readily available in hospitals in our text:

Revised manuscript, line 305-316:

“3.3 Microscopy options for imaging centimeter-sized specimens

Optical sectioning techniques are crucial for obtaining high-quality volumetric images. Techniques such as confocal microscopes, multi-photon microscopy, and light-sheet microscopy filter out-of-focus signals, resulting in sharp images of individual planes. In our study, we used light-sheet microscopy and multi-point confocal (i.e., spinning disc) for imaging centimeter-sized specimens because of their scanning speeds. While two-photon and confocal microscopy offer high-resolution imaging of smaller volumes, they are not ideal for scanning entire tissues because of their prolonged scanning times.”

Non-optical sectioning wide-field fluorescence microscopes, like the Olympus BX series or ZEISS Axio imager series, can also be used to scan samples up to about 3.5mm thick with long working distance objective lenses. In these cases, deconvolution algorithms are required to eliminate out-of-focus signals. However, it should be noted that the epifluorescence system might reduce fluorescent intensity in deeper regions within the samples.”

Refractive index matching is a critical point in the protocol, the one providing final transparency. Authors utilized the commercial solutions NFC1 and NFC2 (Nebulem, Taiwan) with a known refractive index, but for which its composition is non-disclosable. My knowledge on the organic chemistry around refractive index matching is limited, but if users don't really know what is going on in this final step, the whole protocol would rely on a single world-wide provider and troubleshooting would be fishing. I suggest that you try to validate the approach with solutions of known composition, or at least provide the solutions sold by other providers.

We appreciate the reviewer's suggestions. Based on our experience, the CUBIC-R solution developed by Ueda's team also serves as an effective RI-matching solution in the MOCAT pipeline. Its only drawback is the potential reddening of the specimen, likely due to the light-responsive component, antipyrine. We have now added this information to the Methods section:

Revised manuscript, line 492-496:

“Refractive index (RI) matching. Before imaging, the specimens were RI-matched by being immersed in NFC1 (RI = 1.47) and NFC2 (RI = 1.52) solutions (Nebulum, Taipei, Taiwan). Each immersion lasted for one day at room temperature. Alternatively, RI-matching can also be accomplished by immersing specimens in a 1:1 dilution of CUBIC-R[28] for one day, followed by pure CUBIC-R for an additional day.“

**Reviewer #2 (Recommendations For The Authors):**
A comment on the name of the protocol, MOCAT. I am sorry to bring this now, and not before. But, I strongly recommend another name for the procedure. My concern is that the present name "MOCAT" refers to the problem, and NOT to the actual solution provided by you. See, the problem to solve is: to perform Multiplex labeling Of Centimeter-sized Archived Tissue (MOCAT), but it says nothing about HOW you did it: heat-induced antigen retrieval and Tween20-delipidation for centimeter-scale FFPE specimens. In summary, I strongly recommend that the acronym of the procedure refers more to the "solution" than to the "problem", and for me this is important because otherwise the acronym is not fair with present and future techniques pretending to provide a novel solution to the same problem. Another way to put it is that researchers can own their proposed solutions, but they do not own the problem to be solved.

We appreciate the reviewer's suggestions. In response to their concerns, we have renamed the procedure presented in this study as Heat-Induced FFPE-based Tissue Clearing, with the acronym HIF-Clear. This change reflects the critical step in our procedure. Corresponding updates have also been made in the manuscript.